# Weather systems associated with synoptic variability in the moist margin

Corey Robinson[1,2,3], Sugata Narsey[4], Christian Jakob[1,3], and Hanh Nguyen[4]

[1]School of Earth, Atmosphere & Environment, Monash University, Clayton, VIC, Australia
[2]ARC Centre of Excellence for Climate Extremes, Monash University, Clayton, VIC, Australia
[3]ARC Centre of Excellence for 21st Century Weather, Monash University, Clayton, VIC, Australia
[4]Bureau of Meteorology, Melbourne, VIC, Australia

**Correspondence:** Corey Robinson (corey.robinson@monash.edu)

**Abstract.** The moist margin is a sharp gradient of humidity that separates the moist deep tropics from the drier subtropics, and its movement is known to have an important effect on rainfall variability. In this work, we investigate how weather systems are related to synoptic variability in the moist margin. The weather systems considered include convectively coupled equatorial waves and the Madden-Julian Oscillation (MJO), monsoon low-pressure systems (LPS), and extratropical interactions with
the moist margin characterised by upper-level potential vorticity (PV) anomalies. We use an object-based approach in which first, objects are defined to describe the variability of the moist margin, and then are related to weather objects representing the above weather systems. Overall, the results indicate that these weather systems are associated with a large proportion of variability in the moist margin. The MJO and equatorial Rossby waves have a significant modulating effect on the moist margin. In comparison, monsoon LPS are infrequent but strongly influence the moist margin when they occur. Interactions
with the extratropics occur for around one quarter of moist margin perturbations, and display a clear extratropical wave-like signal, often with anticyclonic PV anomalies near the perturbed margin and cyclonic PV anomalies upstream. Overall, moist margin objects associated with weather systems are larger, longer-lived, and precipitate more, highlighting the important role of weather systems.

## 1 Introduction

In the tropics, water vapour and rainfall have been shown to be strongly related (e.g., Sherwood, 1999; Bretherton et al., 2004; Neelin et al., 2009). Observations reveal that the tropical atmosphere is largely characterised by strong variability in water vapour and weak variability in temperature, which has led to the development of various theories in which moisture is a dominant factor controlling precipitation and large-scale dynamics (e.g., Charney, 1963; Neelin and Held, 1987; Sobel et al., 2001; Raymond and Fuchs, 2007; Adames et al., 2019; Mayta and Adames Corraliza, 2024).

Mapes et al. (2018) showed that tropical rainfall at any given time is contained in the region of very high total column water vapour (TCWV). They identified the value of TCWV to represent the threshold below which rainfall is rare, and termed this the moist margin. Robinson et al. (2024b) showed that this moist margin is highly variable on a range of timescales. In particular, they showed strong variability exists at synoptic, or weather, timescales. This begs the question of what the

underlying dynamical processes involved in these perturbations are and whether they are of tropical or extratropical origin (or both).

The tropical atmosphere is home to a rich variety of weather systems that control precipitation, particularly on synoptic and intraseasonal timescales. While pressure gradients are typically weak in the tropics, disturbances and low-pressure systems occasionally form, the strongest of which are tropical cyclones, providing up to around half of the total precipitation depending on the region (Berry et al., 2012; Lavender and Abbs, 2012; Hurley and Boos, 2015). Meanwhile, intraseasonal variability is often related to convectively coupled equatorial waves and the Madden-Julian Oscillation (MJO) (Madden and Julian, 1971). While initially derived from the dry shallow water equations through the work of Matsuno (1966), these waves couple to convection and so moisture are an important aspect of these systems (Wheeler et al., 2000; Kiladis et al., 2009).

Extratropical dynamics can also have an important effect on tropical weather systems, and pose an important challenge for theory and models (Stan et al., 2017). Tropical-extratropical interactions are often characterised by the combination of extratropical waves and tropical moisture (de Vries, 2021), typically involving the poleward transport of moisture and associated enhanced convection (Gimeno et al., 2014; Knippertz, 2007; Todd et al., 2004; Funatsu and Waugh, 2008; Knippertz et al., 2013).

The purpose of this paper is to relate synoptic-scale variations in the moist margin to weather systems in the tropics and extratropics. To do so we define objects for weather systems and perturbations in the moist margin. Many previous studies have developed algorithms to track and identify a variety of weather systems, including but not limited to those presented in Sprenger et al. (2017). One advantage of an object-based approach is that it can lend insight into the dynamics responsible for the system of interest. This is particularly true when objects are tracked through time, allowing for a 'quasi-Lagrangian'[1] analysis of processes that control the development, maintenance and decay of weather systems.

The weather systems investigated in this study include convectively coupled equatorial waves (CCEWs) and the Madden-Julian Oscillation (MJO), monsoon low-pressure systems (LPS), and upper-level potential vorticity (PV) anomalies, with their corresponding object identification algorithms presented in Section 2. Each weather system is expected to have different relationships with the moist margin. For example, CCEWs and the MJO may cause the moist margin to expand or contract based on the phase of the waves, while LPS may be associated with poleward displacements of the margin. The relationship with PV anomalies is likely more complex, since interactions between tropical convection and extratropical waves can go in both directions.

The remainder of the paper is laid out as follows. Section 3 presents results for the interactions between moist margin objects and weather objects, including their frequency, properties and structure. In Section 4, we investigate some temporal aspects of the moist margin by tracking objects and relating these "events" to weather systems through time. A discussion and conclusion are provided in Sections 5 and 6 respectively.

---

[1]This terminology is borrowed from Hauser et al. (2023) and differs from 'Lagrangian' approaches which typically involve trajectory analysis.

## 2 Methods

### 2.1 Data

Our analysis is primarily completed with the ERA5 reanalysis, provided by the European Center for Medium-Range Weather Forecasts (Hersbach et al., 2020). Data for total column water vapour (TCWV) and mean sea level pressure (MSLP), as well as potential vorticity (PV) on the 350 K isentropic level, are resampled to daily means at 1° horizontal resolution over 1979-2021.

Outgoing longwave radiation (OLR) data are taken from the interpolated product from the National Oceanic and Atmospheric Administration (Liebmann and Smith, 1996), which provides daily data at 2.5° horizontal resolution. Daily accumulated rainfall data are provided at 1° resolution by the Global Precipitation Climatology Project (GPCP) version 1.3 (Adler et al., 2020), beginning in October 1996. All analysis involving multiple datasets is performed using the maximum overlap between datasets over 1979-2021, meaning any analysis with rainfall is performed over October 1996 - December 2021.

### 2.2 Object identification

#### 2.2.1 Moist margin perturbations

Perturbations in the moist margin are defined following the methodology in Robinson et al. (2024b) with a TCWV threshold of 45 kg m$^{-2}$. Note that our threshold is slightly different from the 48 kg m$^{-2}$ used in Mapes et al. (2018) due to slight differences between ERA5 and satellite observational datasets. Here we only focus on "wet perturbations", which are defined as areas where the moist margin appears outside its regular background position. More specifically, wet perturbations are areas for which the daily TCWV is above and the "background state" TCWV is below 45 kg m$^{-2}$. Here, the background state is defined as a continuous 90-day running mean, which is different to the 15-day moving-average seasonal climatology used in Robinson et al. (2024b). The reason for this is that the continuous running mean allows for more targeted analysis of "fast" synoptic- to intraseasonal-scale phenomena and largely removes the "slow" seasonal and interannual variability. For example, analysis of the slowly varying component of the moist margin shows substantial correlations with the El Niño Southern Oscillation (not shown), which is not considered here. An example of wet perturbations is shown as the blue regions in Fig. 1, with the background state moist margin at 45 kg m$^{-2}$ denoted by the black contour.

#### 2.2.2 Convectively coupled tropical waves

Convectively coupled equatorial waves and the Madden-Julian Oscillation (MJO), hereafter combined as "convectively coupled tropical waves" (CCTWs), are analysed using the Wheeler-Kiladis spectral decomposition of OLR (Wheeler and Kiladis, 1999) over all longitudes and the latitude band 20°S-20°N. We consider three wave types: the MJO, equatorial Rossby (ER) waves, and Kelvin waves. All other wave types, including mixed Rossby-gravity waves and inertio-gravity waves, are found to be insignificant compared to the three aforementioned wave types and are therefore excluded from our analysis.

The spectral filtering technique requires the choice of zonal wavenumber $k$, period $T$ and equivalent depth $h$ ranges for each wave type. Here we define the MJO with $1 \leq k \leq 5$, $30 \leq T \leq 96$ days and no $h$ condition; the ER wave with $-10 \leq k \leq -1$,

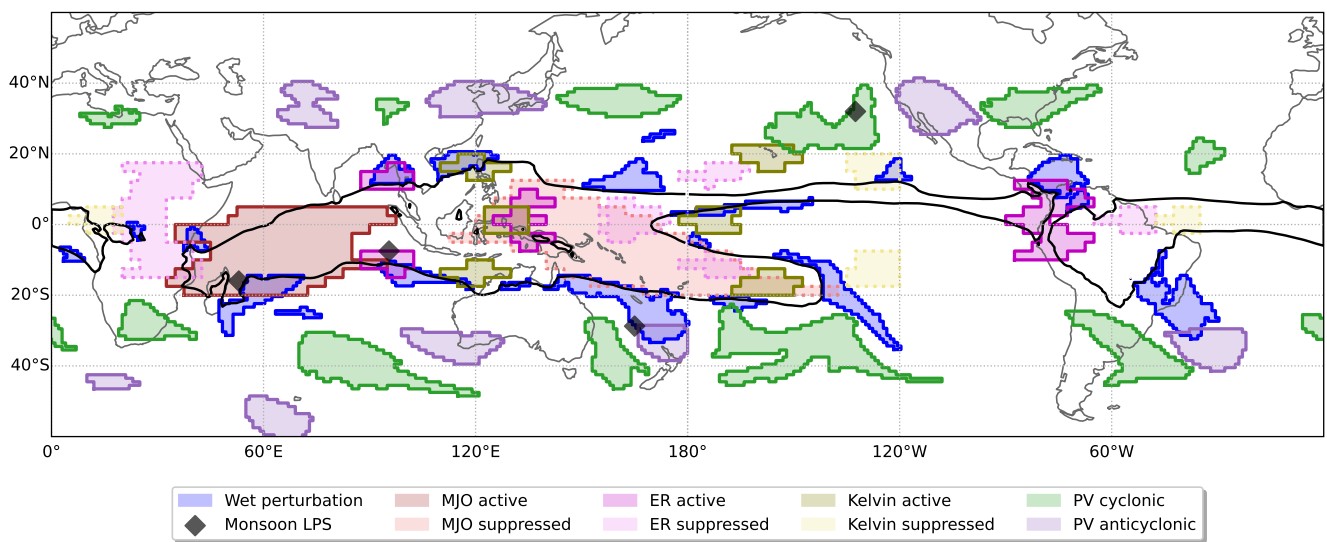

**Figure 1.** Example of tracked objects for 03 January 2018. The background state moist margin, defined as the 45 kg m$^{-2}$ contour of the 90-day running mean TCWV, is in black. Wet perturbations in blue are said to be associated with weather objects if any of the object areas overlap (for MJO, ER and Kelvin objects), or are located within 500 km of the object (for monsoon LPS and PV objects), as described in Section 3.2.

$6 \leq T \leq 80$ days and $8 \leq h \leq 90$ m; and the Kelvin wave with $1 \leq k \leq 14$, $2 \leq T \leq 30$ days and $8 \leq h \leq 90$ m. These values are used operationally by the Bureau of Meteorology in their real-time analysis of CCTWs (http://www.bom.gov.au/climate/enso/#tabs=Tropics), and are largely consistent with typical values used in the literature (Wheeler and Weickmann, 2001; Kiladis et al., 2009). Furthermore, waves must be specified as either symmetric or antisymmetric about the equator. ER and Kelvin waves are symmetric, while there is no symmetry constraint for the MJO.

We are interested in coherent regions where the CCTW substantially modulate convection. Therefore, we define "active" and "suppressed" objects as areas where the filtered OLR anomaly for a particular wave type is below -10 or above +10 W m$^{-2}$ respectively. The threshold of 10 W m$^{-2}$ is chosen since it corresponds to approximately 1 standard deviation for each wave type averaged over the Indian Ocean to West Pacific sector (60-180° E), meaning at any time around 30% of this region will typically be considered either "active" or "suppressed". MJO, ER and Kelvin wave objects can be seen in Fig. 1 in red, pink and yellow shading respectively, with active areas shown in darker colours.

### 2.2.3 Monsoon lows

We use the monsoon low pressure system (LPS) dataset provided by Vishnu et al. (2020), which provides hourly tracks of all low pressure systems in the global tropics (35° N-S). The tracking algorithm is based on the 850 hPa streamfunction from the ERA5 reanalysis, and here the tracks are downsampled to daily temporal resolution for consistency with other data analysed.

The LPS dataset includes a variety of cyclonic phenomena such as monsoon lows and depressions, tropical cyclones, heat lows, and hybrid cyclones, and are denoted by black diamonds in Fig. 1. Because we do not wish to separate between different classifications of low pressure systems, we consider the whole dataset regardless of the properties of the lows. For completeness, we have also repeated the analysis with the tropical cyclone International Best Track Archive for Climate Stewardship (IBTrACS) (Knapp et al., 2010, 2018) and found that all results are qualitatively similar.

### 2.2.4 Potential vorticity anomalies

An effective way of describing synoptic dynamics in the extratropical atmosphere is the analysis of potential vorticity, which is an attractive quantity due to its conservation and invertibility properties, as well as its association with quasi-geostrophic theory which is qualitatively accurate outside the deep tropics (Hoskins et al., 1985). We choose to analyse PV on the 350 K isentropic surface since this level is mostly associated with upper-level Rossby wave dynamics in the lower midlatitudes, and is largely independent of tropical activity such as cyclones. All PV values are multiplied by -1 in the Southern Hemisphere so that positive values are cyclonic everywhere.

Cyclonic and anticyclonic PV anomaly objects are defined as regions where the daily PV is at least 2 PVU (1 PVU = $10^{-6}$ $m^2 \, s^{-1} \, K \, kg^{-1}$) greater or less than the 15-day running mean seasonal climatology respectively. In Fig. 1, these are denoted by green (cyclonic) and purple (anticyclonic) areas. Because we are only interested in the large-scale synoptic distribution of PV, a minimum object size of 250,000 $km^2$ is enforced. This means that some smaller-scale features will be removed. Reducing this threshold results in more objects being detected, but does not significantly change any results or conclusions obtained here (not shown).

## 2.3 Object tracking

In section 4 we present some properties of wet perturbation "events", which requires tracking moist margin perturbations as objects through time. Tracking of wet perturbation objects is performed with the TintX Python package (https://github.com/antarcticrainforest/tintX) introduced in Bergemann et al. (2022), which is an extension to the TINT package previously used to track convective cells in radar data (Raut et al., 2021). This algorithm calculates the phase correlation between consecutive time steps and utilises the Hungarian maximum matching algorithm to generate tracks of object geometries and their centres of mass. Furthermore, the scheme also handles the splitting and merging of objects. More details on the algorithm can be found in Section 2.4 of Bergemann et al. (2022).

## 3 Results

### 3.1 Object frequency

Figure 2 shows the frequency of occurrence of all objects introduced in the previous section, regardless of any interactions between objects. Wet perturbations commonly occur around the shoulder regions of the tropical moist margin, decreasing in

frequency moving away from the climatologically wet regions. By definition, the deep tropics also have a low frequency of wet perturbations due to these regions almost always being above 45 kg m$^{-2}$. For the CCTW objects, we have only presented the frequency of "active" objects since the frequency of "suppressed" objects is identical. This is because filtered OLR anomalies are normally distributed with a mean of zero, meaning values of $\pm 10$ W m$^{-2}$ are equally likely. MJO objects are most common over the Indian Ocean to west Pacific, while ER objects are more common in the west Pacific. Kelvin wave objects are frequent along the equator, particularly in the eastern Hemisphere. Monsoon LPS are overall the least common of the objects used here, occurring less than 10% of the time except for some maxima in the Indian Ocean and Bay of Bengal. By construction, PV anomaly objects are mostly confined to the midlatitudes around 30-50° N/S. Cyclonic anomalies tend to occur slightly equatorward compared to anticyclonic anomalies and can occasionally extend into the deep tropics, highlighting the potential for interactions with the tropical moist margin. Both cyclonic and anticyclonic anomalies are common over the Pacific and Atlantic oceans, and display hints of the "westerly duct" in the central and east Pacific where Rossby waves are known to propagate towards and even across the equator (e.g., Tomas and Webster, 1994)

## 3.2 Weather object interactions with the moist margin

The main objective of this study is to relate variability in the moist margin to weather systems. To do this, we associate each wet perturbation object with one, multiple, or no weather objects. We begin our analysis using a "Eulerian" perspective where each (daily) occurrence of each object is considered to be independent. We will study the temporal evolution of objects in a "Lagrangian" framework in Section 4. For the following discussion, it may be useful to refer to the example of objects shown in Fig. 1.

A wet perturbation is considered to be associated with a CCTW object if there is any overlap between their areas. For example, the wet perturbations centred at around 40°E, 5°S; 60°E, 20°S; and 100°E, 15°S in Fig. 1 all overlap and are hence associated with an MJO active object. For PV anomaly objects, this condition is relaxed to the minimum distance between objects being less than 500 km, to account for PV objects occurring at higher latitudes in recognition of their remote effects (Bishop and Thorpe, 1994). For example, the wet perturbation around 170°E, 30°S in Fig. 1 is associated with both an anticyclonic PV anomaly (which is overlapping) and a cyclonic PV anomaly (which is located less the 500 km to the west). The 500 km distance condition also applies to the point-based monsoon LPS tracks for which we can only assume a characteristic object extent. In Fig. 1, this means that wet perturbations at 60°E, 20°S and 170°E, 30°S are associated with monsoon LPS, while the LPS at 130°W, 30°N is not related to any wet perturbations. The complete set of wet perturbations can then be analysed to identify which weather objects, if any, they are associated with.

We have also tested distance thresholds of 250 and 750 km for PV and LPS objects. While increasing the threshold results in a greater number of wet perturbations being associated with these weather objects (as expected for a more lenient condition), the properties of these interactions are mostly unchanged. We therefore only present results for the 500 km threshold for conciseness.

The proportion of total wet perturbations that are associated with each weather object, defined as the number of associated wet perturbations divided by the total number of wet perturbations, is shown in Fig. 3. Note that here the "CCTW" is the union

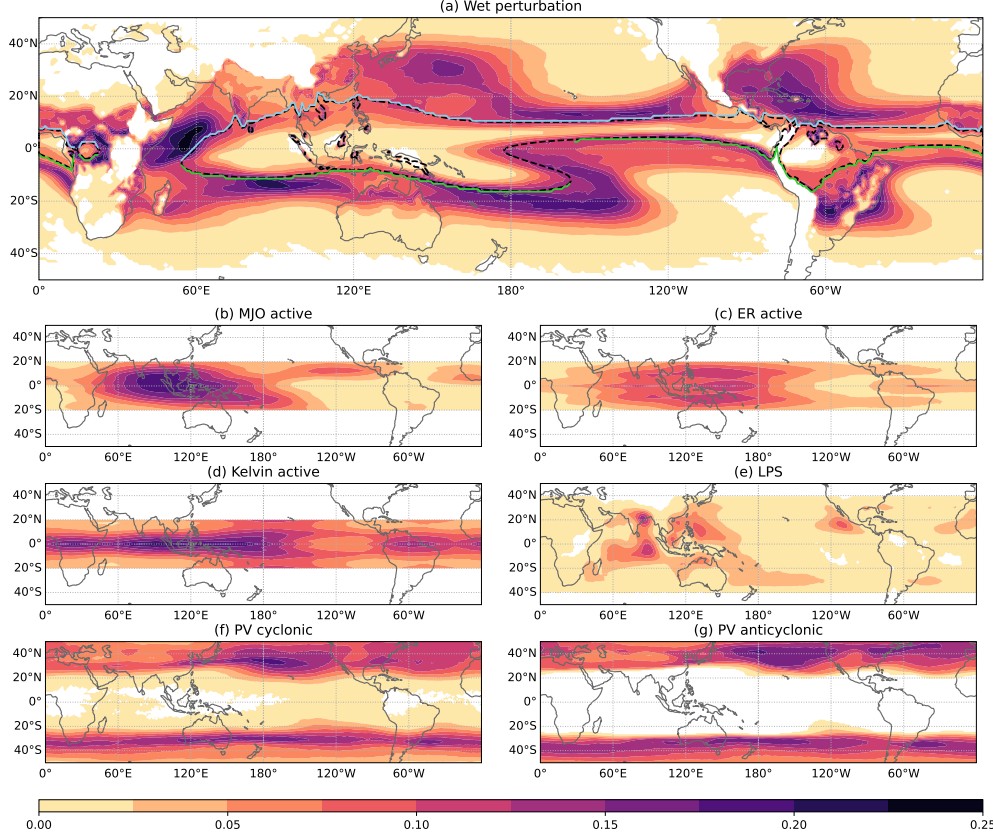

**Figure 2.** Frequency maps of wet perturbations (top panel) and all weather objects analysed (bottom panels). Areas with no objects are in white. In panel (a), the annual-mean moist margin is denoted by the dashed black contour, and its northern and southern edge are marked by blue and green lines respectively. The frequency of the suppressed regions of tropical waves (MJO, ER, Kelvin) are identical to the active phase shown in this figure.

of MJO, ER and Kelvin categories. We also present a category for which no weather objects or only suppressed CCTW objects are associated (termed "none/suppressed"), denoting objects where weather systems are unlikely to be responsible for the wet perturbation.

For all seasons, wet perturbations are frequently associated with the active part of tropical waves (36.7% for any of the 3 types). However, a large number overlap the suppressed part of the wave too (29.1% for any of the 3 types). While Kelvin waves are the most frequently related of the three wave types, there is only a small difference between the active (20.1%) and suppressed (15.9%) phases. In other words, wet perturbations are about 26% more common during active Kelvin wave phases compared to suppressed phases, suggesting Kelvin waves are not a strong predictor for the presence of a wet perturbation. In comparison, both the MJO and ER are slightly better predictors, with wet perturbations being 47% and 55% more common respectively during the active phase compared to the suppressed phase. 24.8% of wet perturbations are associated with PV

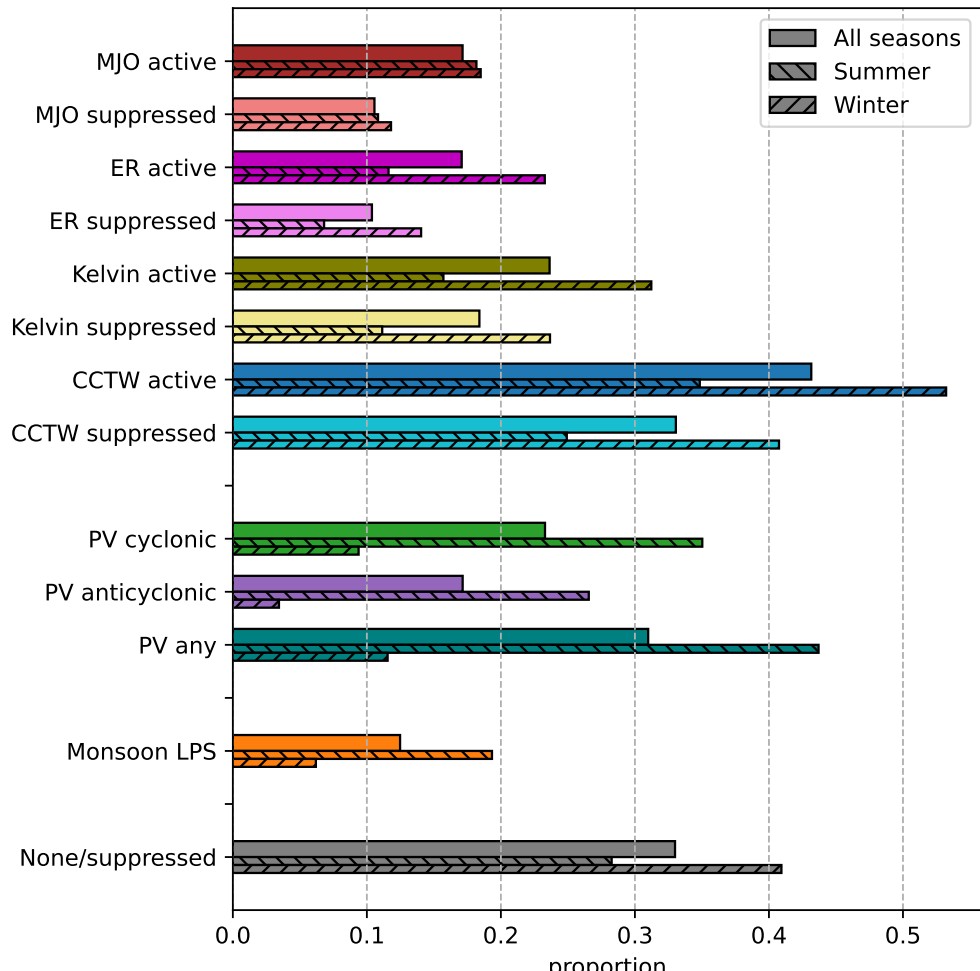

**Figure 3.** Proportion of wet perturbations that are associated with weather objects. Note that proportions sum to a value greater than one since wet perturbations can be associated with multiple weather objects. The CCTW category is the union of MJO, ER and Kelvin categories. "Summer" corresponds to JJA in the Northern Hemisphere and DJF in the Southern Hemisphere, while "winter" corresponds to DJF in the Northern Hemisphere and JJA in the Southern Hemisphere.

175    anomalies of either sign, with slightly more associated with cyclonic anomalies (18.5%) compared to anticyclonic (13.1%). Monsoon LPS are only associated with a small proportion (9.1%) of total wet perturbations. This small number is mainly due to the relative rarity of LPS compared to other objects (e.g. see Fig. 2), and we will see later that LPS are a strong predictor for wet perturbations. Finally, 43% of wet perturbations are not associated with any weather objects or only associated with suppressed CCTW objects.

180      There is also some seasonality in the proportion of wet perturbations associated with weather objects. Monsoon LPS interactions are more likely during summer, consistent with their increased frequency during summer (Hurley and Boos, 2015).

Wet perturbation interactions with PV anomalies are also more common in summer, likely due to the closer proximity of the climatological moist margin to the summer hemisphere jet. In comparison, ER and Kelvin waves are more commonly associated during winter, and the MJO displays weak seasonality. Overall, weather systems are more likely to interact with the moist margin in summer than in winter; this is shown by the lower proportion of "none/suppressed" in summer.

It is also valuable to consider the geographical locations for which wet perturbations are associated with weather objects. Figure 4 shows the maps of proportion of wet perturbations associated with each weather object. Note here that if a wet perturbation object is associated with a particular weather object, the whole object area is considered in the counts. That means wet perturbations extending past 20° N/S may be associated with CCTW objects despite the latter only being defined in 20° N-S.

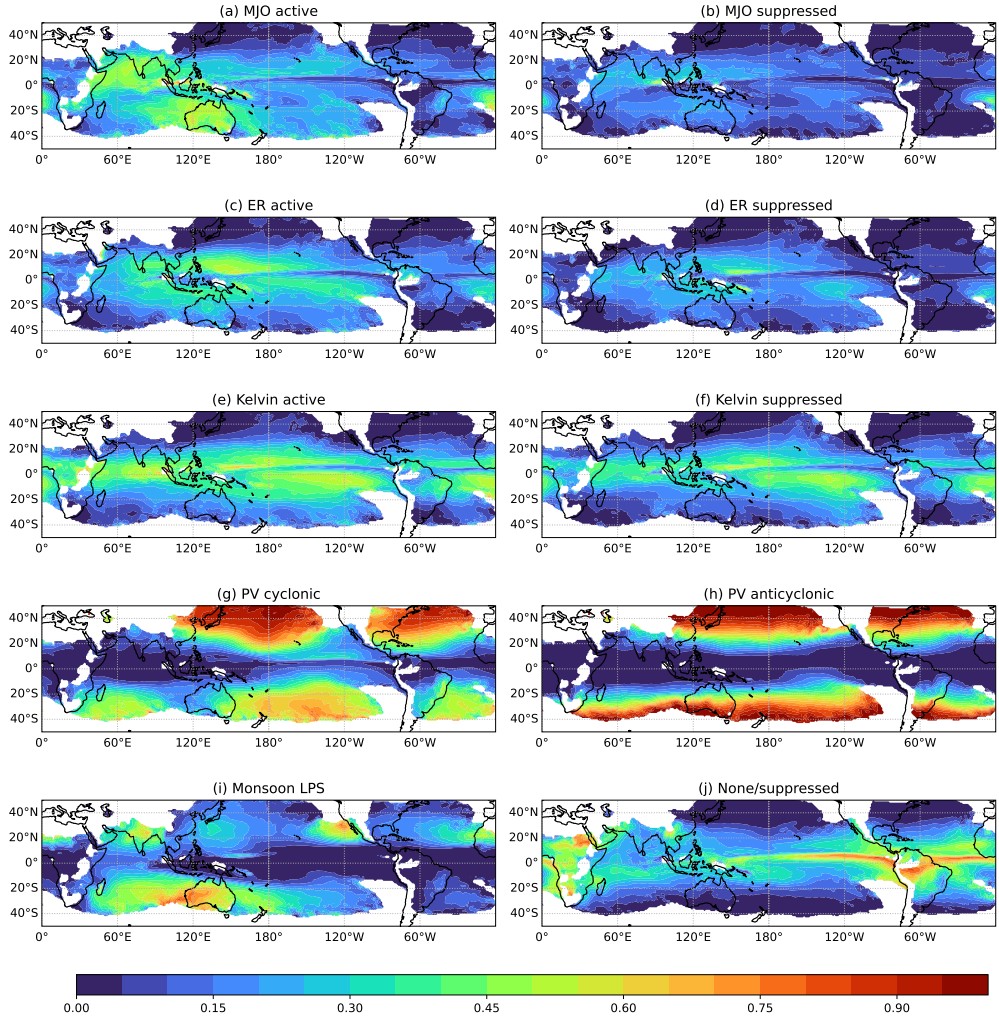

**Figure 4.** Spatial distribution of the proportion of wet perturbations that are associated with weather objects.

Wet perturbations are commonly associated with the active region of the MJO in the off-equatorial Indian Ocean and Australia (up to 50%). The pattern for suppressed MJO objects is similar but weaker overall, suggesting wet perturbations are more likely around the active MJO. Wet perturbations are most commonly associated with ER around the Maritime Continent and West Pacific, and are again more frequent in the active phase compared to the suppressed phase. In comparison, wet perturbations close to the equator, except for some small regions in the east Pacific and Atlantic, are often associated with Kelvin waves; however, there is not much difference between active and suppressed phases.

Wet perturbations that are associated with PV anomalies occur increasingly towards higher latitudes, which is consistent with the frequency of PV objects (Fig. 2f,g). In particular, over 75% of wet perturbations are associated with cyclonic PV anomalies over some regions of the North Pacific, south-central Pacific, and North Atlantic. Meanwhile, almost all wet perturbations extending poleward of around 30° N/S are associated with anticyclonic PV anomalies. There are likely some perturbations that are associated with both cyclonic and anticyclonic PV anomalies given the 500 km distance condition used here. We note that these results indicate that while wet perturbations are relatively rare in the extratropics, when they do occur there, they are very likely associated with a PV anomaly.

Monsoon LPS contribute to a large proportion of wet perturbations in regions of the subtropics such as the Indian Ocean and Australia, India, and the northeast Pacific and Atlantic. Keep in mind that once again, while these proportions are large, the total number of wet perturbations occurring here is small (see Fig. 2a), so these interactions are relatively uncommon.

The wet perturbations that are not associated with any weather object or only with suppressed CCTW objects mostly occur in the deep tropics, close to the equator. This is particularly notable in the east Pacific and Atlantic Oceans, as well as some land areas such as Africa and South America. These perturbations may be related to processes not examined here, as presented later in the discussion.

Overall, Fig. 4 suggests that wet perturbations closer to the equator are likely to be related to CCTW objects, while those further away are more likely associated with LPS and/or PV anomalies, consistent with the frequency patterns of weather objects (Fig. 2).

## 3.3 Significance testing for object interactions

The object-based approach used in this study means that some wet perturbation objects may interact with weather objects by random chance. It is therefore important to consider how the probability of weather systems interacting with wet perturbations (as shown in Figures 3 and 4, for example) differs to these "random chance" interactions. We now describe a statistical test to evaluate whether these probabilities are significantly different.

For each day from 01 Jan 1979 to 31 Dec 2021, wet perturbation objects are compared to weather objects for the same day of year on a randomly chosen year (aside from the same year as the wet perturbations), which can be considered a set of random weather objects for that time of year. The probability of "random chance interactions" is calculated using the same overlap or distance conditions as "true interactions" described in Section 3.2. We then treat each year of data as a sample, resulting in 43 samples of true and random interactions. We apply a two-sided Kolmogorov-Smirnoff test to test the probability of the true and

random samples being drawn from the same population. The test reveals that all weather objects have a statistically significant relationship with the frequency of wet perturbations at a confidence level of 99%.

### 3.4 Characteristics of moist margin perturbations by associated weather system

We now examine some properties of wet perturbations according to their associated weather system. In this section, we have combined the "active" MJO, ER and Kelvin wave objects into a single CCTW category, since results are largely similar between the three wave types. In comparison, the cyclonic PV and anticyclonic PV categories are kept separate due to important differences between the two.

Figure 5 presents the distributions of wet perturbation size, mean TCWV anomaly, and mean daily precipitation for each of the associated weather objects. Wet perturbation area sizes are strongly positively skewed for all categories, indicated by the means being substantially larger than the medians. This implies that there are many small objects and fewer large objects. The distribution is shifted towards larger values for wet perturbations that interact with weather objects, in particular the monsoon LPS and PV anomaly categories. This suggests that perturbations associated with weather systems tend to be larger. Conversely, objects not associated with any weather object (i.e. the "none/suppressed" category) tend to be smaller. Note that, however, this may be partially due to the higher probability of large wet perturbations randomly interacting with weather objects.

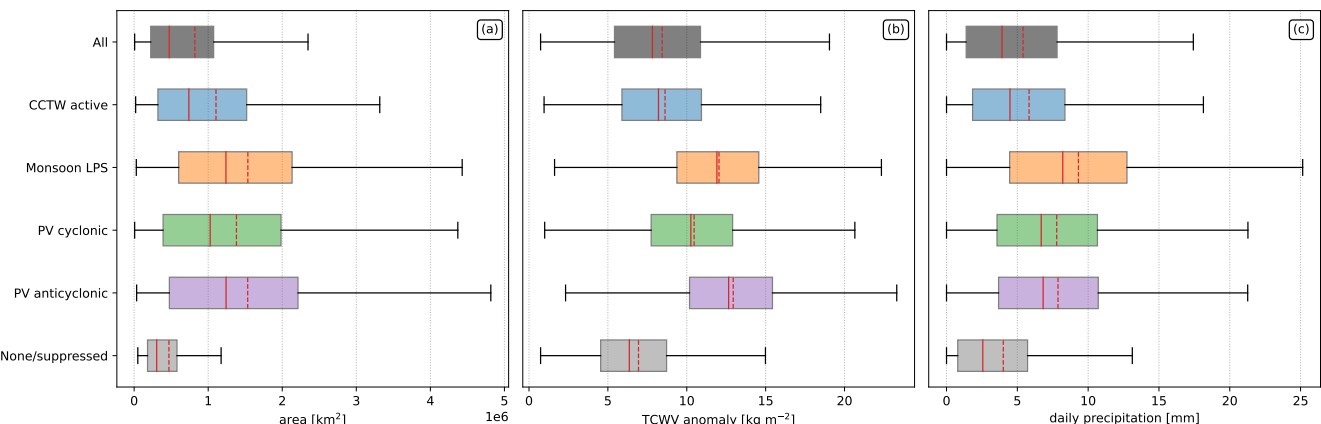

**Figure 5.** Box plots of area (left), mean TCWV anomaly (centre) and mean daily precipitation (right) of tracked wet perturbations that are associated with weather objects. The solid and dashed red lines denote the median and mean of the distributions respectively. Outliers have been removed for the sake of visual clarity.

A similar pattern can be observed for the mean TCWV anomaly of wet perturbation objects. Wet perturbations associated with anticyclonic PV anomalies tend to have the largest TCWV anomalies, followed by monsoon LPS, cyclonic PV anomalies, and active CCTW objects in that order. The order is similar for precipitation, although wet perturbations associated with monsoon LPS precipitate more than those associated with anticyclonic PV anomalies. Once again, objects associated with no weather objects or only suppressed CCTW objects have the weakest TCWV anomalies and the least precipitation.

Further insight can be gained by examining the horizontal structure of wet perturbations that are associated with different weather objects. Figure 6 presents composites of TCWV, precipitation, 350 K PV and MSLP anomalies in the region sur-

245 rounding wet perturbation objects. Composites are presented in pseudo latitude and longitude coordinates where the origin corresponds to the centre of mass of the object. These composites have been taken for all objects that are connected to either the southern or northern edge of the background state moist margin, meaning that some objects that are located in the central Pacific where the background state overturns, as well as any isolated objects, are removed (e.g. see Fig. 1 near the dateline for an example, as well as Fig. 2 for an example of the southern and northern boundaries). This removes objects for which

250 the poleward and equatorward directions are ambiguous. Furthermore, any objects in the Southern Hemisphere are reversed in latitude so that positive pseudo latitude always corresponds to the poleward direction.

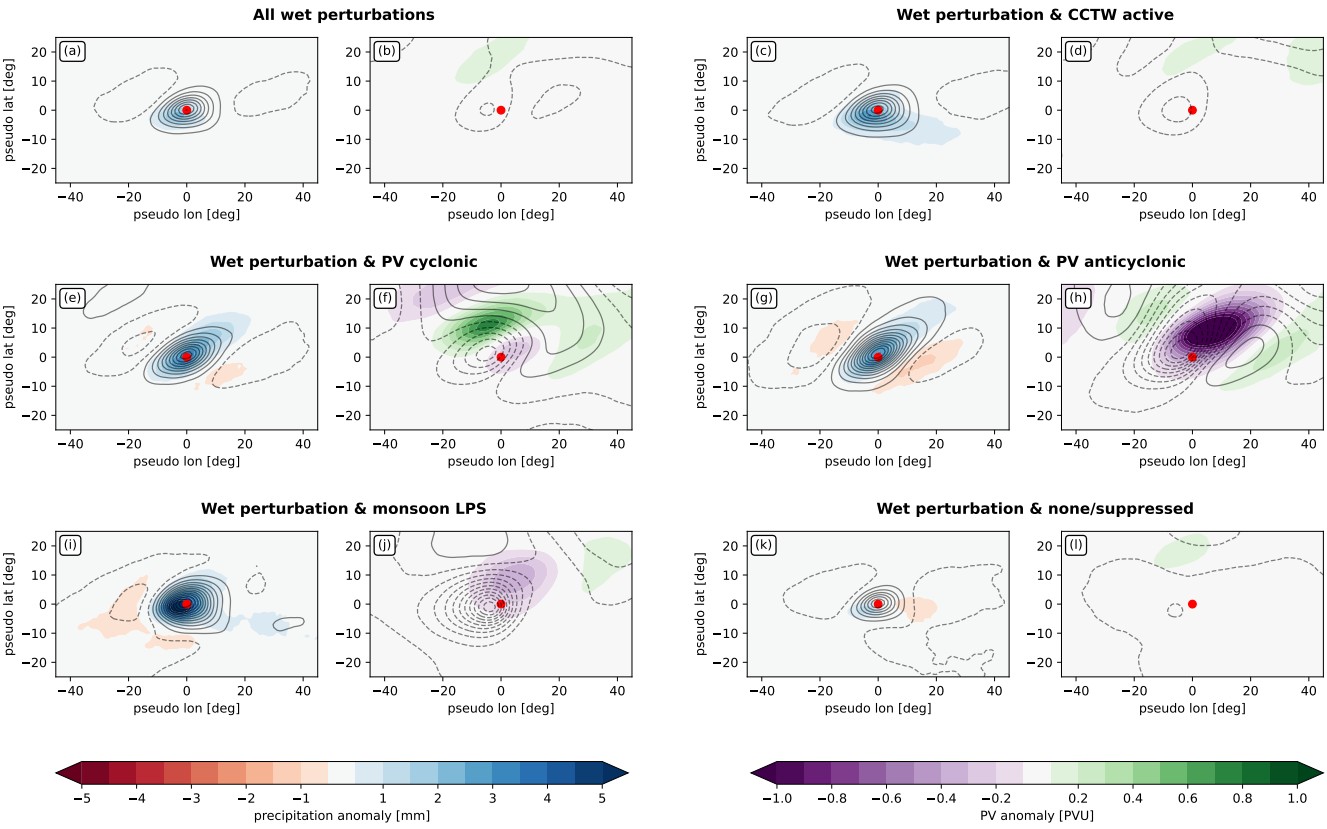

**Figure 6.** Object-relative composites in the region surrounding the centre of mass of wet perturbations (denoted by a red circle) that are associated with weather objects. The left panels for each category show TCWV anomalies (contours, contour interval 1 kg m$^{-2}$) and precipitation anomalies (shading). The right panels show MSLP anomalies (contours, contour interval 0.25 hPa) and PV anomalies (shading). Negative values in contour plots are shown as dashed lines. All objects in the Southern Hemisphere are reversed in latitude so that positive pseudo latitude corresponds to the poleward direction.

In general, wet perturbations tend to be anomalously moist (by construction) and rainy over a longitude and latitude spans of around 15 and 10° respectively, with a slightly dry anomaly to the east and west (Fig. 6a). They are associated with slightly reduced MSLP and cyclonic PV anomalies on the poleward side (Fig. 6b). Wet perturbations associated with the active phase of CCTWs (panels c, d) share a very similar structure that is slightly larger and stronger, particularly in the rainfall which shows positive anomalies extending more than 20° eastward.

In comparison, wet perturbations related to PV anomalies (Fig. 6e-h) show a substantially different structure. There exists a wave-like pattern in the moisture that slants eastward and equatorward for both cyclonic and anticyclonic PV anomalies, along with a prominent maximum in precipitation extending poleward and eastward. Furthermore, both TCWV and precipitation anomalies are larger than the cases of all wet perturbations and those related to CCTWs. The wave-like pattern is also reflected in the dynamical structure, although here there are some differences between cyclonic and anticyclonic PV anomalies. Wet perturbations associated with cyclonic PV anomalies (panels e, f) display the cyclonic PV anomaly poleward and westward of the object centre, as well as an isolated surface low located equatorward of an anomalous surface ridge, implying poleward and eastward flow and moisture transport into higher latitudes. Meanwhile, wet perturbations associated with anticyclonic PV anomalies (panels g, h) display the anticyclonic PV anomaly located poleward and eastward of the object centre, with a strong low-pressure trough extending equatorward and westward to poleward and eastward.

Wet perturbations associated with monsoon LPS (Fig. 6i,j) also have stronger anomalies than average. In particular, there are strong positive TCWV and precipitation anomalies and negative MSLP anomalies over an area of approximately 20x20°, and a weak anticyclonic PV anomaly slightly poleward and eastward. Finally, wet perturbations unrelated to any weather object or only to suppressed CCTWs (Fig. 6k,l) show the weakest anomalies of all categories. Overall, the results presented in Fig. 5 and 6 suggest that weather systems play an important role in determining the properties of wet perturbations and their associated rainfall, particularly monsoon LPS and PV anomalies.

## 3.5   Effect of weather systems on the moist margin

The approach in Section 3.2 was to consider all wet perturbation objects and associate them with a variety of weather objects. However, the association of moist margin perturbations to the likelihood of co-occurring weather objects may be confounded by the relative frequency of each weather object itself. As an extreme hypothetical example, let us consider a weather object which exists at all times in all locations. Although the analysis in the previous section would indicate that all wet perturbations are related to this weather object, in reality this provides no information on what role the weather object plays on the moist margin. Therefore, this section aims to establish the conditional probability in the opposite direction – what is the likelihood of moist margin perturbations in the presence of a weather object?

The results of this analysis are presented in Fig. 7, which shows odds ratios for the occurrence of wet perturbations relative to the centre of mass of each weather object. Once again, objects that are located in the Southern Hemisphere are reversed in latitude so that the poleward direction is represented by positive pseudo latitude. The odds ratio is defined as the probability of a wet perturbation existing given the weather objects exists divided by the base probability of a wet perturbation existing. More specifically, for each instance of a weather object we compare the current presence of a wet perturbation to the climatological

frequency of wet perturbations (as in Fig. 2a) in the surrounding region. This means that the odds ratios account for the spatially varying climatology of objects. Odds ratio values above (shown in green) and below one (shown in purple) mean that there is an increased or decreased chance of a wet perturbation existing respectively.

Wet perturbations are more frequent over a wide area surrounding MJO active objects (Fig. 7a). The maximum odds ratio exceeds 1.5 near the centre of the MJO object, and this signal extends poleward and eastward. Meanwhile, the presence of a suppressed MJO object largely decreases the likelihood of wet perturbations (panel b), with odds ratios below 0.5 near the centre of the objects. The odds ratios for ER wave objects (panels c, d) is of a similar magnitude, but the structure is more confined, consisting of wet-dry patterns of zonal wavelength around 50-60°. This is consistent with the smaller scale of ER waves compared to the MJO. In comparison, the odds ratios for Kelvin waves (panels e, f) are closer to 1 and more isolated, suggesting that Kelvin waves are a weaker predictor for wet perturbations than the MJO or ER waves (consistent with results in Section 3.2).

Wet perturbations are much less likely near the centre of cyclonic PV objects, but more likely poleward and westward, as well as equatorward and eastward of the object (Fig. 7g). Meanwhile, wet perturbations are much more likely near the centre of anticyclonic PV objects, and less likely to the east and west (panel h). Both cyclonic and anticyclonic PV anomalies show an equatorward-refracting wave pattern, which is consistent with the composites shown in Fig. 6e-h. For monsoon LPS, wet perturbations are more likely to the east and less likely to the west of the low (panel i). This is consistent with poleward and equatorward moisture transport on the eastern and western flanks of the LPS respectively. It is also apparent that LPS and PV objects have a stronger effect on modulating the frequency of wet perturbations than CCTW objects, with odds ratios further from 1.

## 3.6 Summary of object analysis

In this section, we have investigated the relationship between perturbations of the moist margin, defined through wet perturbation objects, with a variety of weather systems. Carrying out the analysis in both directions (i.e. assigning perturbations to weather objects, and conditional probabilities of perturbations in the presence of weather objects) helped to describe the role each weather systems plays in moist margin variations in a statistical sense.

CCTWs (MJO, ER and Kelvin waves) are associated with a large proportion of variability in the moist margin. Although Kelvin waves are associated with the most wet perturbations (Fig. 3), this is only because they are the most frequent object, and in fact they have the least predictive power out of the three wave types (Fig. 7). We conclude that CCTWs have a modulating effect on moist margin variability, but are only one of many factors related to variability of the moist margin.

Monsoon LPS are only associated with a small proportion of wet perturbations, but this is mainly because they are infrequent compared to other objects analysed. Both the composite structure (Fig. 6i,j) and odds ratios (Fig. 7i) suggest that LPS are strongly related to variability in the moist margin when they occur.

Interactions of the moist margin with the extratropics can be characterised by proximity to large-scale PV anomalies, both cyclonic and anticyclonic. Indeed, Fig. 4g,h shows that wet perturbations extending far poleward almost always interact with PV anomalies of either sign. Note that unlike CCTWs, there is no a priori assumption for cyclonic and anticyclonic PV anoma-

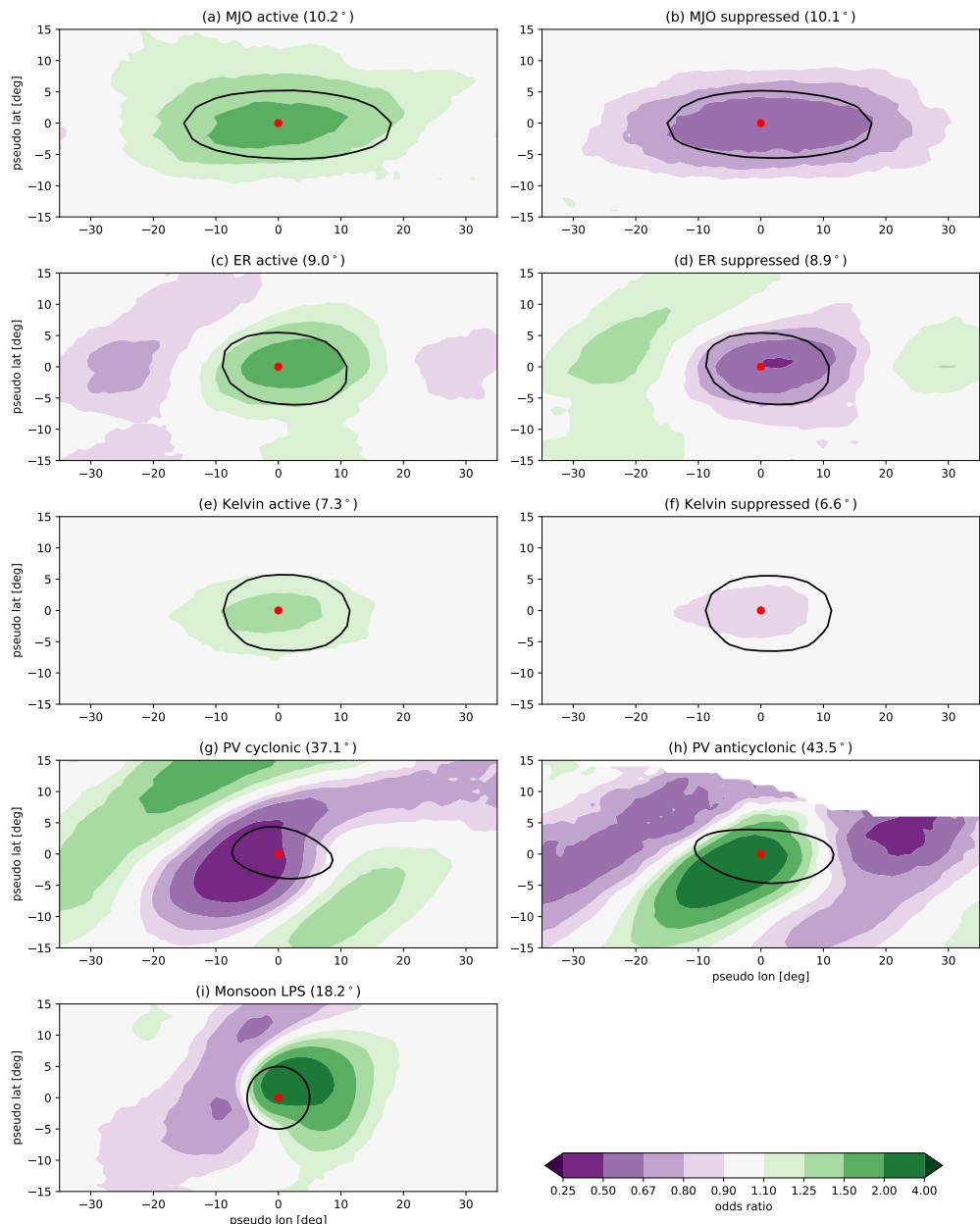

**Figure 7.** Odds ratios for the occurrence of wet perturbations relative to the centre of mass of tracked weather objects (denoted by a red circle). Positive pseudo latitude corresponds to the poleward direction, and negative to equatorward. The grey contour displays the average size and shape of objects denoted by 50% of weather object occurrence relative to that object's centre, except for monsoon LPS which are defined with a 500 km radius. Note the nonlinear scale. Areas where wet perturbation frequency is less than 0.05% have been masked out. The number in the panel headings corresponds to the average absolute latitude of weather objects.

lies having opposite relationships with the moist margin; therefore we consider both objects as potentially being important for the moist margin. In particular, cyclonic and anticyclonic PV anomalies are closely related to each other through the trough-ridge structure of extratropical Rossby wave trains, meaning the same fundamental processes are responsible for both objects (Wirth et al., 2018). Wet perturbations associated with PV anomalies display strong composite anomalies that cover a large meridional extent compared to other object types. Further, the presence of PV objects strongly modulates the frequency of wet

perturbations (Fig. 7g,h).

It is possible that multiple weather systems combine for a greater effect on the moist margin. This is particularly relevant for CCTWs, since the active region of the MJO is often itself associated with ER and Kelvin waves (Kiladis et al., 2005). Investigating this reveals that the average rainfall of wet perturbations increases (decreases) when interacting with mutiple active (suppressed) CCTW objects, although this is a relatively small effect and each weather system contributes largely

independently (not shown).

A key result in this section is that wet perturbation objects not associated with any weather systems tend to be the least important (i.e., they are the smallest, have the weakest TCWV anomalies, and produce the least amount of rainfall), highlighting the importance of weather systems in controlling the moist margin. However, these relationships may partially exist due to smaller wet perturbations being less likely to be associated with weather systems by random chance.

## 4   Event-based analysis of the moist margin

One limitation of the approach taken in the previous section is that each instant (day) of a wet perturbation object is considered to be independent of every other. Also, the weather objects and wet perturbation objects are only matched instantaneously. Therefore, it is difficult to attribute any cause and effect between the moist margin and weather systems. To gain some first insights into the temporal aspects of the relationships, we seek to move from a 'Eulerian' object-based framework to a "La-

grangian" event-based framework, tracking objects through time (see Section 2.3 for details).

### 4.1   Event properties by weather system

Object tracking allows us to investigate transient properties of wet perturbation events, such as event movement and lifetime. Some properties of wet perturbation events associated with weather systems are presented in Fig. 8. Note that here, a wet perturbation event is said to be associated with a weather object if the conditions in the previous section are satisfied (i.e.

overlapping for CCTWs, and within 500 km for monsoon LPS and PV anomalies) at any point throughout the duration of the event, with each event only counted once. Zonal and meridional propagation velocities of the wet perturbations are calculated as an average throughout the event duration by taking the difference between ending and starting centres of mass and dividing by the event duration. Because this requires at least two points, only events at least two days long are considered for the velocity calculations. Furthermore, positive meridional velocities correspond to the poleward direction in both hemispheres.

Figure 8a shows the distribution of the maximum poleward latitude throughout the duration of each wet perturbation event. The bulk of wet perturbation events extend to around 10-22.5° N/S, though some extend well poleward of this. It is evident that

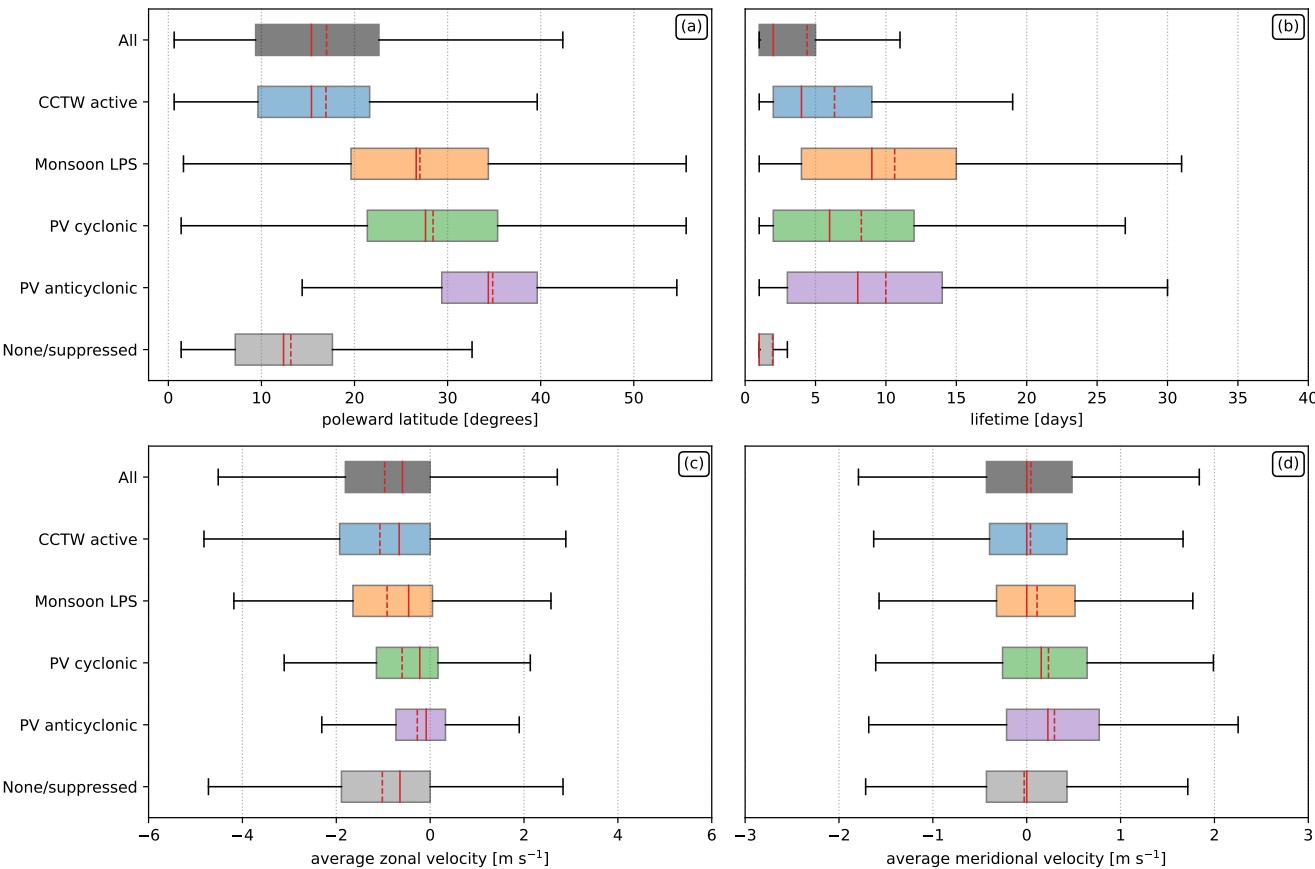

**Figure 8.** Boxplots of most poleward latitude (a), lifetime (b), average zonal velocity (c) and average meridional velocity (d) of wet perturbation events by associated weather systems. Positive meridional velocities correspond to the poleward direction in both hemispheres. The solid and dashed red lines denote the median and mean of the distributions respectively. Outliers have been removed for the sake of visual clarity.

events that extend towards the extratropics are associated with monsoon LPS and/or PV anomalies, in particular anticyclonic PV anomalies. In comparison, events that are never associated with any weather object or only with suppressed CCTW objects tend to remain in the tropics.

The lifetime of wet perturbation events is positively skewed with many short events and fewer long events. In particular, events unrelated or only related to suppressed CCTWs tend to be very short-lived with a median of only one day. Meanwhile, events related to monsoon LPS tend to live the longest (median of 13 days), followed by anticyclonic and cyclonic PV anomalies (median of 9 and 8 days respectively) and then CCTWs (median of 4 days).

Wet perturbations tend to travel westward at an average of around 1 m s$^{-1}$, although there is a large spread. Approximately 360    three-quarters of events travel westward. Events that are associated with CCTWs, monsoon LPS, or no weather objects have

similar distributions. However, events associated with cyclonic or anticyclonic PV anomalies have a more even distribution with nearly half travelling eastward. These results are consistent with the dominant wind patterns which shift from easterlies in the tropics to westerlies in the midlatitudes (as seen in panel a, events associated with PV tend to be located further poleward). However, a large proportion of these still travel westward; this may be due to issues with splitting and merging of objects, as well as the condition that the interaction with weather objects only has to be satisfied for one day. The meridional velocities of wet perturbations are overall smaller than the zonal velocities, and travel poleward and equatorward approximately equally. This is likely due to the geometry of the moist margin, which covers a large zonal extent but is confined meridionally. However, events associated with PV anomalies and monsoon LPS tend to travel poleward slightly more often.

## 4.2 Weather systems through the life cycle of a wet perturbation

The results in the previous section indicate that wet perturbations that extend towards the extratropics tend to be related to PV anomalies, and to a lesser extent, monsoon LPS. However, it is unclear to what extent these weather objects are related through the life cycle of wet perturbation events. For example, does a wet perturbation begin with the active part of a CCTW and then grow poleward after interacting with a PV anomaly?

To investigate this further requires the definition and analysis of an "event life cycle" by identifying the start and end of an event as well as finding its "peak". The start and end are simply defined as the first and last days of the tracked object. For the event peak, we choose the day when the object has its largest meridional extent. To ensure a coherent event life cycle is detected, we only consider events that are at least three days long. This removes around 56% of events, although, as seen in Fig. 8b, these are mostly events not associated with any weather object.

Figure 9 presents the proportion of wet perturbation events that are associated with weather objects throughout the life cycle of wet perturbation events. These have been separated into events that do and do not extend poleward of 22.5°, which is the 75th percentile of the maximum poleward latitude distribution (as in Fig. 8a).

There are very clear differences in the interactions between the moist margin and weather objects depending on the wet perturbation stays close to the deep tropics or moves further poleward. Objects that remain within tropical latitudes start out without any interaction with weather objects around 60% of the time and through interactions with CCTWs around one third of the time. In contrast, about 60% of the wet perturbations that move further poleward are already associated with weather objects at the start.

This pattern remains throughout the entire duration of the event, with objects that extend poleward more commonly associated with monsoon LPS and PV anomalies (but less commonly with CCTWs) for both the event peak and end. Likewise, LPS and PV anomalies are only infrequently associated with wet perturbations that remain in the deep tropics. Wet perturbations, regardless of whether they move poleward or not, are more commonly associated with weather objects of any kind during the peak of the event compared to the start or the end. Furthermore, almost all events lasting at least three days interact with at least one weather system at any point throughout their lifetime, which is seen by the small grey bars in the "any time" category. This is particularly true for objects that move poleward, with 64% related to active CCTWs, 43% to monsoon LPS, 71% to cyclonic PV anomalies, and 58% to anticyclonic PV anomalies (recall that an event may be related to multiple weather objects).

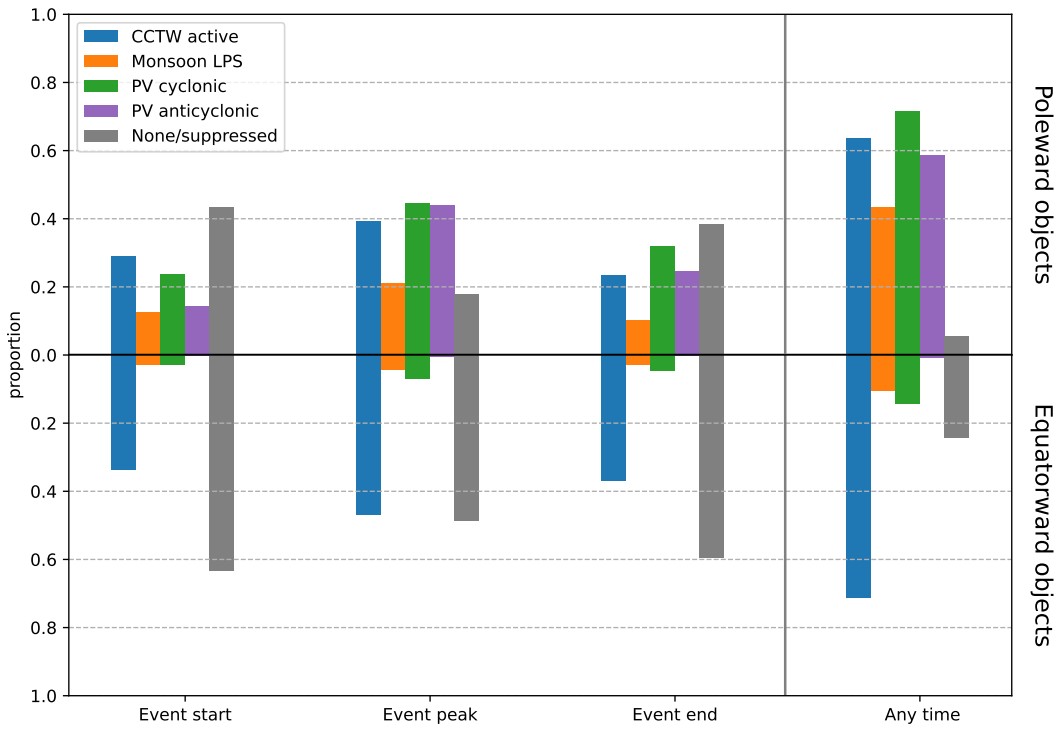

**Figure 9.** Proportion of wet perturbation events associated with weather objects throughout their life cycle, for events that last at least three days. The right-most charts shows the proportion of events that are associated at any time throughout the event, as in Fig. 8. Events that extend poleward of 22.5° are plotting in upward bars, and those that do not in downward bars.

## 5 Discussion

We have related variability of the tropical moist margin to a variety of weather systems, including convectively coupled tropical waves, monsoon low-pressure systems, and upper-level potential vorticity anomalies. Poleward movement of the moist margin is described through "wet perturbation" objects, which are then related to weather objects. Results are consistent between the Eulerian object-based framework and the Lagrangian event-based framework. Overall, our results indicate that these weather systems are all associated with but have different relationships with the moist margin. We now discuss the role of each weather system in the following paragraphs.

Convectively coupled tropical waves (CCTWs), including the Madden-Julian Oscillation, equatorial Rossby waves and Kelvin waves, are associated with a large amount of variability in the moist margin, especially in the deep tropics. Wet perturbations are more (less) likely to occur around the convectively active (suppressed) area of a CCTW, demonstrating the importance of these waves. However, this effect is somewhat limited, especially for Kelvin waves, with wet perturbations also frequently observed around the suppressed part of the wave. Furthermore, wet perturbations associated with CCTWs tend to exhibit fairly average characteristics (such as size and rainfall), perhaps due to their high prevalence. Therefore, we conclude

that CCTWs have a slight "modulating" effect on the moist margin depending on their phase. This is similar to previous literature describing the modulation of precipitation by CCTWs (e.g., Wheeler et al., 2000; Kiladis et al., 2009; Lubis and Jacobi, 2015; van der Linden et al., 2016; Muhammad et al., 2024). Since CCTWs typically have their largest amplitudes in the deep tropics and within the moist margin (e.g., see Fig. 5 of Kiladis et al., 2009), it is possible that CCTWs, in particular Kelvin waves, have a stronger effect on variability within the moist margin rather than along the margin itself. CCTWs may also have an indirect effect on variability at higher latitudes through interactions with the subtropical jet; this is discussed further below with respect to PV anomalies.

Monsoon low-pressure systems (LPS) are found to be infrequent, but have an important effect on the moist margin when they occur. This is particularly true in the subtropics, where a large proportion of wet perturbations are associated with them. Wet perturbations associated with monsoon LPS tend to be larger, longer-lived, and rain more than normal. Our results also suggest that LPS are important contributors to the movement of moisture (and associated precipitation) from the tropics into the subtropics, which is consistent with previous literature (Hurley and Boos, 2015; Schumacher and Galarneau, 2012; Robinson et al., 2024a).

When the moist margin extends beyond around $25°$ poleward, it is commonly associated with upper-level potential vorticity (PV) anomalies. This is perhaps expected given that sequences of cyclonic and anticyclonic PV anomalies are a characteristic feature of the extratropics. However, our results suggest that this connection extends beyond mere coincidence. Wet perturbations associated with PV anomalies tend to be large with strong TCWV anomalies, produce abundant rainfall, and have a strong dynamical structure. Furthermore, the presence of PV anomalies greatly alters the odds of wet perturbations occurring (Fig. 7), suggesting they have a strong influence on the moist margin. In particular, wet perturbations appear to be strongly linked with anticyclonic PV anomalies and tend to co-occur (Fig. 1, 7), and this interaction sometimes exists at the start of wet perturbation events that move poleward (Fig. 9). It is plausible that these interactions are associated with upper-level divergence above a convectively active region, with the divergent wind acting to generate anticyclonic PV anomalies through its interactions with the subtropical jet (Parker et al., 2013; O'Brien and Reeder, 2017; Teubler and Riemer, 2021). Such a process may indicate an indirect effect of CCTWs on PV anomalies extending beyond the convectively active region of the wave. A more detailed and process-oriented examination of interactions between upper-level PV and the moist margin is the subject of ongoing work.

A significant amount of variability (43% of wet perturbations) in the moist margin has not been linked to any of the weather systems in this study. However, our results indicate that these are the smallest and shortest-lived events, and produce the least amount of precipitation. The results in Fig. 8 suggest that moist margin events lasting at least three days often begin and end with no associated weather system, although it is unlikely that there are no weather systems related throughout the whole event lifetime. This suggests that many perturbations in the moist margin may begin from small quasi-random oscillations that are not connected to any synoptic-scale systems analysed here, and instead are perhaps related to local and mesoscale processes. They may also be related to moisture modes, which are hypothesised to grow from an instability along a strong gradient of TCWV (Adames Corraliza and Mayta, 2024; Mayta and Adames Corraliza, 2024). For the perturbations to grow, synoptic-scale dynamics become more important. It is also important to note that we have not considered an exhaustive list of synoptic weather systems in this study, which may account for some of these perturbations. One example is the African Easterly wave,

which may be important for generating wet perturbations over Africa and the Atlantic Ocean (Thorncroft and Hodges, 2001). Moist margin perturbations may also be related to midlatitude fronts and troughs (e.g., Narsey et al., 2017), although it is likely that these are also related to PV anomalies and therefore will not be part of the 'none' category.

## 5.1 Limitations

We recognise that there are limitations with the object-based approach used in this study. In particular, defining objects requires the use of numerous thresholds which can be somewhat arbitrary. Sensitivity testing for some quantities have been carried out as described in Section 2 and are found to not have any significant changes to our results. Another important limitation is that it is difficult to determine any cause and effect between weather systems and the moist margin. For example, does the weather system perturb the moist margin, or does the perturbed moist margin result in a weather system? While the odds ratio plots in Fig. 7 and "Lagrangian" analysis in Fig. 9 go some way toward this, further analysis is needed to confidently ascribe any causality of weather systems on the moist margin.

Other limitations include the restricted domain of 20° N-S for the CCTW dataset with wet perturbations commonly existing poleward of these latitudes. However, note that our condition for interactions with the moist margin only requires part of the objects to overlap, and the climatological moist margin tends to remain within 20° N-S. Therefore, we do not expect the domain extent to have a major impact on results. The interpretation of the moist margin is also limited over land regions and in the higher latitudes (especially in the winter hemisphere); these have been discussed previously in Robinson et al. (2024b). Because moist margin perturbations are more difficult to attribute to weather objects in regions such as Africa and South America, future work could investigate what weather systems not considered here may be responsible for variability in the moist margin in these regions.

## 6 Conclusion

This work has shown that variability in the moist margin is closely related to synoptic weather systems such as tropical waves, tropical lows and cyclones, and upper-level potential vorticity anomalies. Moist margin perturbations that are connected to weather systems are more important (i.e. they tend to be larger, last longer, and have stronger TCWV anomalies and rainfall), demonstrating the significance of these weather systems and their associated synoptic dynamics. In particular, around one quarter of perturbations are associated with potential vorticity anomalies, and these tend to extend further poleward, demonstrating their interaction with the extratropics. The interaction of poleward moving wet perturbations with weather systems remains different throughout its life cycle when compared to events that remain in the deep tropics. Future work on this topic will further examine the relationship and mechanisms between PV anomalies and the moist margin.

*Code and data availability.* All datasets used in this study are publicly available online. ERA5 reanalysis data are available from the ECMWF Copernicus Climate Data Store (Hersbach et al., 2020). Outgoing longwave radiation data are provided by the National Oceanic and Atmo-

spheric Administration (Liebmann and Smith, 1996). GPCP rainfall data was received from the NCAR Research Data Archive (Adler et al., 2020). The monsoon low pressure system dataset is available on Zenodo at https://doi.org/10.5281/zenodo.3890646.

*Author contributions.* CR performed the formal analysis and prepared the original manuscript with contributions from all co-authors. HN calculated the filtered outgoing longwave radiation anomalies used in the study. SN and CJ provided important guidance and feedback throughout the project.

*Competing interests.* The authors declare that they have no conflict of interest.

*Acknowledgements.* This research is supported by the Australian Research Council Centre of Excellence for Climate Extremes (CE170100023), the ARC Centre of Excellence for the Weather of the 21st Century (CE230100012), and the ARC Discovery Project (DP200102954). We acknowledge the National Computational Infrastructure for their provision of computational resources and services. We are grateful to Andries de Vries and an anonymous reviewer for providing insightful comments, as well as Matthew Wheeler and Acacia Pepler for useful discussions, all of which greatly helped improve the manuscript.

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
