# Peer review of "Weather systems associated with synoptic variability in the moist margin"

_EGUsphere, 2024_

## Referee Comment (RC1)

**Review of manuscript "egusphere-2024-3150"**

Title: Weather systems associated with synoptic variability in the moist margin

Authors: Corey Robinson, Sugata Narsey, Christian Jakob, and Hanh Nguyen

**Summary and recommendation**

The study investigates the association of synoptic-scale weather systems with transient structures of enhanced atmospheric water vapor in regions just outside the very moist tropical atmosphere. The authors investigated the role of equatorial waves, monsoon lows, and upper-tropospheric potential vorticity anomalies in modulating the occurrences and characteristics of these moist air perturbations. They found a spatially varying relevance of these weather systems in the occurrence and properties of wet perturbations as can be expected from the different atmospheric circulation regimes from tropical and extratropical origin.

I found the manuscript very well written, clearly structured, presenting very interesting results based on a thorough and solid analysis. The research goals, methods, findings, and conclusions are in line with another. I very much appreciate the clear presentation and description of the results. I recommend publishing the manuscript after the authors considered two general comments and several very minor comments as detailed below.

**General comments**

1. Relevance of moist margin objects associated with weather systems.
While the overall analysis of the study is very interesting, carefully designed, and informative, my only more general comment is that the information presented in Figures 3 and 4, and the corresponding text in section 3.2, does not account for the spatially varying climatology of the considered weather systems. For example, more than 35% of wet perturbations is associated with active CCTWs (Fig. 3), and around 60% of wet perturbations at 180E, 20N are linked to cyclonic PV anomalies (Fig. 4g). This information doesn't necessarily tell much since these fractions may arise from the matching of wet perturbation and weather system objects by chance. In other words, the fractions may be strongly influenced by the underlying, spatially varying climatology of the wet perturbations and weather systems, as the authors also recognize in lines 231-238. To evaluate the importance/association of the weather systems for/with wet perturbations, the authors may consider to add some form of significance testing (for example, based on a Monte Carlo test) or by computing odds ratios of wet perturbations occurring in association with the weather systems with respect to the matching of these objects if they would occur completely independent of another (for example, by comparing the observed fractions of wet perturbations associated with weather systems to the median fractions of wet perturbations associated with weather systems based on randomized timeseries from a Monte Carlo test).

2. Seasonality.
As mentioned, in my opinion, the analysis is very thorough and leads to interesting and meaningful results. That saying, the study is based on a year-round analysis, and I was wondering if the authors had considered looking into seasonality. For example, are there strong deviations in the relationship between the wet perturbations and weather systems during different seasons of the year? This is merely meant as a suggestion to consider since the study is already rather complete as is.

**Specific comments**

Line 5. Please, consider writing "upper-level potential vorticity (PV) anomalies" for more accuracy?

Line 8. "equatorial Rossby waves"?

Line 10. To clarify the extratropical nature, perhaps write, "extratropical wave-like signal"?

Line 11. Consider replacing "remotely" by "upstream" (as derived from Fig. 6f)?

Line 11. Specify the text speaks about "moist margin objects"?

Line 30. Perhaps write "derived from the dry shallow water equations"?

Line 96. Would it be useful specifying the focus is on the "lower midlatitudes" instead of the more general "extratropics"?

Line 102. The authors may consider removing the phrase "such as PV streamers" since such streamers can easily exceed the mentioned surface area threshold.

Section 2.3. It may be helpful for readers to add one-two sentences how the tracking of objects in time is realized, for example, based on spatial overlap of the objects in daily mean fields?

Line 118. The phrase "are very nearly normally distributed" reads somewhat odd to me and rephrasing may be considered.

Figure 2a. Would it be possible to add some form of reference of the climatological moist margin, or moist tropical reservoir, for example, by adding a contour(s) where at least 50% of the days of the year the TCWV is larger than 45 kg m-2 or simply the annual or seasonal mean TCWV contour at 45 kg m-2?

Lines 155-156. Given that the text speaks about weather systems as "predictors" of moist margin objects, some form of significance testing or odds ratios in spatial maps may be valuable given that the weather systems objects have a (strong) spatially different climatology and may "by chance" be linked to moist margin objects. Please, see general comment #1.

Lines 184-188. While the description of the results is accurate, some interpretation may be added. For example, does the relatively high wet perturbation frequency in some of regions near the equator tell that mesoscale processes are dominating the occurrences of wet perturbations in these regions? Also, the increasing relevance of cyclonic and anticyclonic PV anomalies towards higher latitudes seems to reflect the latitudinal dependency of the extratropical forcing.

Lines 198-199. A word of caution may be added here. The larger association of wet perturbations with weather systems, and vice versa, may (in part) stem from the higher likelihood of large wet perturbations overlapping by chance with any weather system compared with smaller wet perturbations.

Line 200-202. The described order of different weather systems linked to wet perturbation properties seems to be correct for the precipitation (Fig. 5c) but not for TCWV anomaly (Fig. 5b) where anticyclonic PV anomalies are associated with largest TCWV anomalies, followed by LPS and cyclonic PV anomalies.

Line 206-207. Could this "southern or northern edge of the background state moist margin" be added to Fig. 2a (please, see also one of the previous specific comments). In addition, it may be helpful for readers adding the motivation for the choice removing specific structures.

Figure 6. I would be curious to know how the composites are constructed given the spatially varying distances between grid points across latitudes.

Line 217. The "wave-like pattern" seems to emerge in the moisture, PV, and MSLP fields, but not necessarily in the "precipitation" field?

Line 221. Perhaps add some interpretation on the observed "cyclonic PV anomaly poleward and westward of the object center" implying a poleward and eastward flow transporting moisture of tropical origin into higher latitudes.

Line 241-242. Do the defined odds ratios also account for the spatially varying climatology of the weather systems and wet perturbations? In other words, if let's say LPS are relatively frequent in a specific part of the world, are the relative changes of nearby wet perturbations considered for these specific regions?

Line 249. Did the authors perhaps mean to say "more zonally confined"?

Lines 275-276. Perhaps some interpretation can be added on the cyclonic and anticyclonic PV anomalies which although independently defined, are in reality closely associated with another, being part of extratropical Rossby wave trains (with embedded trough and ridge structures) propagating into low latitudes (as visible in Fig. 1 weat and east of Australia and over South America).

Lines 280-281. "tend to be the least important", perhaps specify in terms of surface area, moisture and precipitation properties? In addition, as mentioned before, a word of caution may be added since larger wet perturbations have by definition a larger likelihood to be connected with weather system objects.

Line 306. Likely a typo, "m s-1".

Figure 8. What surprises me quite a bit is that the zonal velocity of wet perturbations associated with cyclonic and anticyclonic PV anomalies is primarily westward. I would have expected a clear eastward velocity for the larger part of the distribution given the prevalent westward-eastward direction of extratropical Rossby waves and associated PV anomalies. Do the authors have any explanation in this regard? Could it be there are some artefacts or limitations in the tracking of wet perturbations, missing wet perturbations moving fast in time in extratropical latitudes where background winds are typically larger than in (sub)tropical latitudes, potentially leading to a lack in spatial overlap of structures in daily mean fields (assuming this is how the tracking is defined)?

Line 380. The same reference is mentioned twice.

Line 406. "demonstrating the significance of synoptic dynamics", for what? Please, consider to be more specific.

Line 407. I can't follow exactly the phrase "which occurs around one quarter of the time"; does it refer to wet perturbations associated with the PV anomalies or to the climatology of the PV anomalies itself? Instead, would a more general interpretation of the role of the extratropical forcing be valuable?

---

## Referee Comment (RC2)

This paper presents a study into the variability of the moist margin, a boundary in the tropics where total column water vapour (TCWV) falls below 48 kg m$^{-2}$ (though 45 kg m$^{-2}$ is used in this study). Within this region, there is frequent rainfall. This work builds on recent studies to relate synoptic-scale variations in the moist margin to modes of tropical rainfall variability.

The study utilises an object-based approach, whereby overlaps between different 'weather' objects, such as convectively coupled tropical waves (CCTWs) and wet perturbations of the moist margin, are examined.

In general, the study provides interesting insights into the relationship between different tropical modes and the variation in the moist margin. The following general comments outline ways the manuscript could be improved: the methodology could be explained further, the study could be better motivated in the introduction, and a more expanded discussion of the role of CCTWs in modulating the moist margin could be provided. The manuscript is recommended for publication after the following major and minor comments are addressed.

**Major Comments**

**1. Introduction Expansion**

The introduction is a bit short. It could be expanded to include the following: (a) An explanation of why an object-based approach was used (e.g., whether this methodology has been used in previous studies) and the advantages of using Eulerian and Lagrangian approaches. (b) A discussion of how different modes of tropical variability might influence the moist margin. For instance, it is mentioned in the discussion that potential vorticity (PV) anomalies may be a response to changes in the tropical margin. As tropical heating anomalies influence the extratropics, providing this background earlier would enhance the context.

**2. Clarification and Explanation in Figure 1**
   a. A significant amount of information is provided in the caption of Figure 1. Key details, such as the definition of the moist margin and the association of wet perturbations with weather objects, should be incorporated into the main text.
   b. The section could benefit from a clearer explanation of what is depicted in Figure 1, such as explicitly stating how the MJO active object overlaps with the wet perturbation object and similar interactions for other objects.

c.  Figure 1: The caption states, "The background moist margin, defined as the 90-day running mean of 45 kg m$^{-2}$, is in black," whereas the main text mentions, "Wet perturbations are areas where the daily TCWV is above and where the background TCWV is below 45 kg m$^{-2}$." This appears contradictory. Clarify whether there is a distinction between the background moist margin and the background state, and define these terms in the text.

**3.  Role of CCTWs in Wet Perturbations**

The discussion does not adequately explain why CCTWs only slightly modulate wet perturbations despite their significance in tropical rainfall variability. One possible explanation is that CCTWs have their highest amplitudes within the moist margin (e.g., see Figure 5 of Kiladis et al., 2009). Kelvin waves, in particular, have their greatest amplitude confined close to the equator, well within the moist margin. Thus, it is plausible that CCTWs primarily contribute to internal variability within the moist margin.

**4.  Interactions of Multiple Weather Objects**

The manuscript does not address cases where more than one weather object contributes to a wet perturbation. For example, Kelvin waves and ER waves are often embedded in active MJO events, forming part of a larger weather event. Are such events double-counted in Figure 6 (panels c and d)? A figure similar to Figure 4 could be included to show events where MJO + Kelvin waves, MJO + ER waves, ER + Kelvin waves, and MJO + ER + Kelvin waves are associated with a wet perturbation. It could be valuable to test the hypothesis that simultaneous modes result in a greater effect on wet perturbations.

**5.  Indirect Influence of CCTWs on the Moist Margin**

As noted in Comment 3, CCTWs are potentially more important for the internal variability of TCWV within the moist margin. However, since they are responsible for high-amplitude rainfall in the tropics, they may release significant latent heat that causes upper-level divergence. This divergence could interact with the subtropical jet, producing PV anomalies. Hence the influence of CCTWs could be indirect. Investigating this could be a point of future work.

**Minor Comments**

**1.  Introduction**

a. Line 15: Consider rephrasing to "Observations reveal that the tropical atmosphere is largely characterised by..."

2. **Section 2.2.2: Convectively Coupled Tropical Waves**
   a. Line 67: Clarify what is meant by the "20-degree latitude band." Does this refer to all latitudes within 20°S-20°N? If so, why does the Kelvin wave contour extend beyond 20°N?

3. **Section 2.3**
   a. Line 107: The URL is not enclosed in brackets.

4. **Section 3.1: Results**
   a. Clarify whether the objects in Figure 2 are only those overlapping with wet perturbations or all identified objects. If the latter, how does this change when considering only objects overlapping with wet perturbations?

5. **Section 3.2**
   a. Figure 3: Define "proportion." Is it the ratio of objects during wet perturbations? Note in the caption that proportions add to a value greater than one, possibly due to multiple objects contributing to a wet perturbation (related to Major Comment 4).
   b. Figure 4: The caption mentions numbers in brackets, but these are absent from the panels.

6. **Section 3.3**
   a. Figure 6: The caption needs clarity. Could the authors explain how the composites are "relative" to the centre of mass and what "by their associated weather objects" means? Also, clarify "pseudo-latitude" and its necessity. Perhaps replace it with "poleward" for better comprehension. Additionally, explain why negative pseudo-latitude is used and what "pseudo-longitude" means in the caption.
   b. Line 216: The text states, "There exists a wave-like pattern that propagates equatorward towards the east." As this is a spatial composite, propagation cannot be inferred. Perhaps rephrase to indicate the pattern slants eastward.

7. **Section 4.1**
   a. Line 290: "A wet perturbation event is said to be associated with a weather object if the conditions in the previous section are satisfied at any point..." Restate these conditions for clarity.

---

## Author Comment (AC1)

We would like to thank both reviewers for their thoughtful and constructive feedback. The comments provided have been greatly valuable and we believe have made substantial improvements to the manuscript. Please see below for our responses to all comments.

All line numbers refer to the tracked changes version of manuscript.

**Response to RC1:**

**Review of manuscript "egusphere-2024-3150"**

Title: Weather systems associated with synoptic variability in the moist margin

Authors: Corey Robinson, Sugata Narsey, Christian Jakob, and Hanh Nguyen

**Summary and recommendation**

The study investigates the association of synoptic-scale weather systems with transient structures of enhanced atmospheric water vapor in regions just outside the very moist tropical atmosphere. The authors investigated the role of equatorial waves, monsoon lows, and upper-tropospheric potential vorticity anomalies in modulating the occurrences and characteristics of these moist air perturbations. They found a spatially varying relevance of these weather systems in the occurrence and properties of wet perturbations as can be expected from the different atmospheric circulation regimes from tropical and extratropical origin.

I found the manuscript very well written, clearly structured, presenting very interesting results based on a thorough and solid analysis. The research goals, methods, findings, and conclusions are in line with another. I very much appreciate the clear presentation and description of the results. I recommend publishing the manuscript after the authors considered two general comments and several very minor comments as detailed below.

**General comments**

1. Relevance of moist margin objects associated with weather systems.
While the overall analysis of the study is very interesting, carefully designed, and informative, my only more general comment is that the information presented in Figures 3 and 4, and the corresponding text in section 3.2, does not account for the spatially varying climatology of the considered weather systems. For example, more than 35% of wet perturbations is associated with active CCTWs (Fig. 3), and around 60% of wet perturbations at 180E, 20N are linked to cyclonic PV anomalies (Fig. 4g). This information doesn't necessarily tell much since these fractions may arise from the matching of wet perturbation and weather system objects by chance. In other words, the fractions may be strongly influenced by the underlying, spatially varying climatology of the wet perturbations and weather systems, as the authors also recognize in lines 231-238. To evaluate the importance/association of the weather systems for/with wet perturbations, the authors may consider to add some form of significance testing (for example, based on a Monte Carlo test) or by computing odds ratios of wet perturbations occurring in association with the weather systems with respect to the matching of these objects if they would occur completely independent of another (for example, by comparing the observed fractions of wet perturbations associated with weather systems to the median fractions of wet perturbations associated with weather systems based on randomized timeseries from a Monte

Carlo test).

We agree that there are some limitations with the interpretation of results from Figures 3 and 4, particularly considering the spatial varying climatology of weather objects. These have been pointed out in lines 278-284 and justify the use of odds ratios in the Section 3.5 to determine the effect weather systems have on the moist margin. However, the analysis could be made more rigorous using statistical tests in Section 3.2 as suggested by the reviewer.

Creating a set of "random weather objects" a la Monte-Carlo methods is challenging, considering the analysis is performed on an object-by-object basis rather than by grid point. We have decided on the following approach to calculate the probability of weather objects interacting "by random chance".

For each day from 01 Jan 1979 to 31 Dec 2021, we calculate the distance between wet perturbations on that day and weather objects on that day of year for a randomly chosen other year. So, for example, wet perturbations on 01 Jan 1979 may be compared to weather objects on 01 Jan 1980, 1981, or any single year up to 2021. The same day of year is chosen to ensure the effects of seasonality are considered in the calculations. This is repeated every day for the 43-year period to create an effectively independent sample of weather objects which preserves object features such as their shape and size distributions. The probability of an interaction "by random chance" can be calculated in the same way as the probability of a "true interaction", except we compare wet perturbations to weather objects in a different year.

To perform a significance test of whether the number of interactions is more or less than what would be observed by random chance, we use the above method to create a sample distribution of probability of interactions with each year as a sample (43 samples total). To save computational costs, we only consider the $5^{th}$, $15^{th}$ and $25^{th}$ days of each month – testing shows results are largely insensitive to the sample size. We then perform a two-sided Kolmogorov-Smirnoff test comparing the distributions of "random interactions" to "true interactions". The results of this test for each weather object types are shown in Figure A1. The p-values are all very small, indicating that the null hypothesis of "wet perturbations are only related to weather systems by chance" can be rejected (at a confidence level of >99% for all objects), meaning weather systems have a significant interaction with the moist margin.

[Figure]

*Figure A1: Histograms of the proportion of "random interactions" (blue bars) with "true interactions" (red bars) per year for each weather object. The p-value of a two-sided Kolmogorov-Smirnoff test is displayed within each panel.*

The description and results of the statistical test have been added into a new subsection (3.3, lines 215-226) in the manuscript.

2. Seasonality.

As mentioned, in my opinion, the analysis is very thorough and leads to interesting and meaningful results. That saying, the study is based on a year-round analysis, and I was wondering if the authors had considered looking into seasonality. For example, are there strong deviations in the relationship between the wet perturbations and weather systems during different seasons of the year? This is merely meant as a suggestion to consider since the study is already rather complete as is.

We agree that the seasonality of the moist margin and its interactions with weather systems are an underexplored aspect of this paper. One may expect that the frequency of weather objects and/or their co-occurrence with moist margin perturbations may depend on the season. To explore this further, we have revised Figure 3 (see Fig. A2 below) to include probabilities for the summer and winter hemispheres (DJF and JJA for the appropriate hemispheres), along with the year-round probabilities.

[Figure]

Figure A2: new version of Figure 3 showing seasonality of wet perturbation interactions with weather systems.

The key result is that interactions with monsoon LPS and PV anomalies are more common during summer. Again, this is likely due to LPS being more common in summer, and the summer edge of the moist margin being closer to the jet and associated PV anomalies. ER and Kelvin wave interactions are more common in the winter hemisphere (perhaps because the edge of the moist margin is closer to the equator), while MJO remains approximately equal throughout the seasons. The proportion of "none/supressed" is somewhat higher in winter.

This figure has been updated in the manuscript and a paragraph has been added describing the effects of seasonality in lines 182-187. However, we have decided to keep the remainder of the analysis as year-round for the sake of conciseness.

**Specific comments**

Line 5. Please, consider writing "upper-level potential vorticity (PV) anomalies" for more accuracy?
Updated

Line 8. "equatorial Rossby waves"?
Updated

Line 10. To clarify the extratropical nature, perhaps write, "extratropical wave-like signal"?
Updated

Line 11. Consider replacing "remotely" by "upstream" (as derived from Fig. 6f)?

Updated

Line 11. Specify the text speaks about "moist margin objects"?
Updated

Line 30. Perhaps write "derived from the dry shallow water equations"?
Updated

Line 96. Would it be useful specifying the focus is on the "lower midlatitudes" instead of the more general "extratropics"?
Updated

Line 102. The authors may consider removing the phrase "such as PV streamers" since such streamers can easily exceed the mentioned surface area threshold.
Updated

Section 2.3. It may be helpful for readers to add one-two sentences how the tracking of objects in time is realized, for example, based on spatial overlap of the objects in daily mean fields?
A sentence has been added here stating that the algorithm uses phase correlations and the Hungarian maximum matching algorithm to generate tracks (lines 124-126).

Line 118. The phrase "are very nearly normally distributed" reads somewhat odd to me and rephrasing may be considered.
The sentence has been rephrased to "…OLR anomalies are normally distributed…" which should be clearer.

Figure 2a. Would it be possible to add some form of reference of the climatological moist margin, or moist tropical reservoir, for example, by adding a contour(s) where at least 50% of the days of the year the TCWV is larger than 45 kg m-2 or simply the annual or seasonal mean TCWV contour at 45 kg m2?
The annual mean moist margin (TCWV = 45 kg m-2) has been added to Figure 2a. Note that there is substantial seasonal variability in the position of the moist tropical reservoir (e.g. see Fig 2 of Robinson et al. 2024, JGR Atmospheres), however given the figure shows annual frequencies we have opted to display the annual mean rather than seasonal means.

Lines 155-156. Given that the text speaks about weather systems as "predictors" of moist margin objects, some form of significance testing or odds ratios in spatial maps may be valuable given that the weather systems objects have a (strong) spatially different climatology and may "by chance" be linked to moist margin objects. Please, see general comment #1.

Odds ratios maps are provided in Figure 7, and significance testing has been performed in response to general comment #1.

Lines 184-188. While the description of the results is accurate, some interpretation may be added. For example, does the relatively high wet perturbation frequency in some of regions near the equator tell that mesoscale processes are dominating the occurrences of wet perturbations in these regions? Also, the increasing relevance of cyclonic and anticyclonic PV anomalies towards higher latitudes seems to reflect the latitudinal dependency of the extratropical forcing.

Mesoscale processes for perturbations in the deep tropics: this has been mentioned in the discussion (lines 442-443). We have included a brief statement and reference to the discussion in lines 211-212.

Relevance of PV anomalies by latitude: we have added a phrase saying the patterns are consistent with the frequency of weather objects (lines 214-215).

Lines 198-199. A word of caution may be added here. The larger association of wet perturbations with weather systems, and vice versa, may (in part) stem from the higher likelihood of large wet perturbations overlapping by chance with any weather system compared with smaller wet perturbations.
We agree with this comment and have added a sentence at the end of the paragraph as suggested (lines 238-239).

Line 200-202. The described order of different weather systems linked to wet perturbation properties seems to be correct for the precipitation (Fig. 5c) but not for TCWV anomaly (Fig. 5b) where anticyclonic PV anomalies are associated with largest TCWV anomalies, followed by LPS and cyclonic PV anomalies.
This has been corrected and rewritten.

Line 206-207. Could this "southern or northern edge of the background state moist margin" be added to Fig. 2a (please, see also one of the previous specific comments). In addition, it may be helpful for readers adding the motivation for the choice removing specific structures.
As well as adding the annual mean moist margin to Figure 2a, we have also added lines denoting the southern or northern edge. Note the discontinuity in the southern boundary in the central Pacific, as described in the text. Also, a sentence has been added in the body text describing why this has been done (lines 252-253).

Figure 6. I would be curious to know how the composites are constructed given the spatially varying distances between grid points across latitudes.
The composites (and odds ratios in Figure 7) are constructed on a regular lat/lon grid. This means that the horizontal spacing of grid coordinates in physical space ($\Delta x$) may be inconsistent since objects may occur at different latitudes. We do not expect this to have any major effect on the composites since most wet perturbations (particularly their centroids), tend to occur at lower latitudes where the curvature of the earth is smaller.

Line 217. The "wave-like pattern" seems to emerge in the moisture, PV, and MSLP fields, but not necessarily in the "precipitation" field?
The lack of a wave pattern in precipitation is partially due to the choice of contours (weak dry anomalies can be seen to the east/west when reducing the contour levels but are still less clear compared to other variables.)

We have revised the text so that precipitation is no longer described as "wave-like" but instead as an "prominent maximum" (line 262).

Line 221. Perhaps add some interpretation on the observed "cyclonic PV anomaly poleward and westward of the object center" implying a poleward and eastward flow transporting moisture of tropical origin into higher latitudes.
This has been added at the end of the sentence.

Line 241-242. Do the defined odds ratios also account for the spatially varying climatology of the weather systems and wet perturbations? In other words, if let's say LPS are relatively frequent in a specific part of the world, are the relative changes of nearby wet perturbations considered for these specific regions?

The odds ratios are defined as the ratio of "conditional frequency" to "climatological frequency". To calculate the "climatological frequency", for each weather object we take the wet perturbation frequency (as in Fig. 2a) in a region surrounding the centre of the weather object, then average over all objects. For the "conditional frequency" the same calculation is performed but instead using the binary mask for wet perturbations on that day. In this sense we are accounting for the spatial variations in the *wet perturbation* climatology. The spatially varying climatology of *weather objects* is also considered since this method iterates over each weather object.

We have added a sentence describing the method in more detail and explaining it accounts for spatially varying climatologies in lines 289-291.

Line 249. Did the authors perhaps mean to say "more zonally confined"?
We have opted for neither "zonally" nor "meridionally" since either word may cause confusion. The direction is clear since "zonal wavelength" is referred to later in the sentence.

Lines 275-276. Perhaps some interpretation can be added on the cyclonic and anticyclonic PV anomalies which although independently defined, are in reality closely associated with another, being part of extratropical Rossby wave trains (with embedded trough and ridge structures) propagating into low latitudes (as visible in Fig. 1 weat and east of Australia and over South America).
We have added a sentence in lines 314-316 stating that these objects are due to the same fundamental processes and are closely linked with each other.

Lines 280-281. "tend to be the least important", perhaps specify in terms of surface area, moisture and precipitation properties? In addition, as mentioned before, a word of caution may be added since larger wet perturbations have by definition a larger likelihood to be connected with weather system objects.
We have clarified that "least important" as having the smallest areas, lowest TCWV anomalies and least amount of rainfall in line 336. The issue of bias towards larger Is mentioned is lines 338-339.

Line 306. Likely a typo, "m s-1".
Fixed

Figure 8. What surprises me quite a bit is that the zonal velocity of wet perturbations associated with cyclonic and anticyclonic PV anomalies is primarily westward. I would have expected a clear eastward velocity for the larger part of the distribution given the prevalent westward-eastward direction of extratropical Rossby waves and associated PV anomalies. Do the authors have any explanation in this regard? Could it be there are some artefacts or limitations in the tracking of wet perturbations, missing wet perturbations moving fast in time in extratropical latitudes where background winds are typically larger than in (sub)tropical latitudes, potentially leading to a lack in spatial overlap of structures in daily mean fields (assuming this is how the tracking is defined)?

There are a few explanations for this. Perhaps most notably, there are limitations associated with tracking wet perturbations and calculating their speeds, especially with regards to the splitting and merging of objects, which may add some noise. There are also instances where a long-lived wet perturbation may only briefly be associated with a PV object, meaning its net movement over its lifetime may still be westward. Finally, as the reviewer as suggested, wet perturbation objects are likely quite transient and do not exist as long in the extratropics due to the stronger westerly winds.

Despite this, wet perturbations associated with PV objects still display a noticeable shift towards more eastward velocities, which we expect is due to the background westerlies in higher latitudes.

We have added a sentence in lines 369-370 providing an explanation for why a large proportion of wet perturbations associated with PV anomalies travel westward.

Line 380. The same reference is mentioned twice.
These are two companion papers with similar titles and the same authors (but reversed in order), which may be causing confusion.

Line 406. "demonstrating the significance of synoptic dynamics", for what? Please, consider to be more specific.
This has been changed to "significance of these weather systems and their associated synoptic dynamics".

Line 407. I can't follow exactly the phrase "which occurs around one quarter of the time"; does it refer to wet perturbations associated with the PV anomalies or to the climatology of the PV anomalies itself? Instead, would a more general interpretation of the role of the extratropical forcing be valuable?
One quarter of wet perturbations are related to PV anomalies – we agree this was not clear in the original text, and as such has been rewritten.

Note we are hesitant to call this "extratropically forced" since the direction of forcing between PV anomalies and the moist margin is still unclear at this stage, and instead we opt for the more general "interact with the extratropics".

**References:**

Robinson, C. M., S. Narsey and C. Jakob (2024). "Synoptic Variability in the Tropical Oceanic Moist Margin." Journal of Geophysical Research: Atmospheres **129**(11).

Wirth, V., M. Riemer, E. K. M. Chang and O. Martius (2018). "Rossby Wave Packets on the Midlatitude Waveguide—A Review." Monthly Weather Review **146**(7): 1965-2001.

**Response to RC2:**

This paper presents a study into the variability of the moist margin, a boundary in the tropics where total column water vapour (TCWV) falls below 48 kg m$^{-2}$ (though 45 kg m$^{-2}$ is used in this study). Within this region, there is frequent rainfall. This work builds on recent studies to relate synoptic-scale variations in the moist margin to modes of tropical rainfall variability.

The study utilises an object-based approach, whereby overlaps between different 'weather' objects, such as convectively coupled tropical waves (CCTWs) and wet perturbations of the moist margin, are examined.

In general, the study provides interesting insights into the relationship between different tropical modes and the variation in the moist margin. The following general comments outline ways the manuscript could be improved: the methodology could be explained further, the study could be better motivated in the introduction, and a more expanded discussion of the role of CCTWs in modulating the moist margin could be provided. The manuscript is recommended for publication after the following major and minor comments are addressed.

**Major Comments**

**1. Introduction Expansion**

The introduction is a bit short. It could be expanded to include the following: (a) An explanation of why an object-based approach was used (e.g., whether this methodology has been used in previous studies) and the advantages of using Eulerian and Lagrangian approaches. (b) A discussion of how different modes of tropical variability might influence the moist margin. For instance, it is mentioned in the discussion that potential vorticity (PV) anomalies may be a response to changes in the tropical margin. As tropical heating anomalies influence the extratropics, providing this background earlier would enhance the context.

We agree that the introduction could be expanded to provide for context for the study.

  (a) We have provided a summary of object-based approaches and their advantages in lines 40-43.
  (b) We have introduced the weather systems analysed in the study and outlined a hypothesis for their effects on the margin in lines 44-50. In particular, we note that the relationship with PV anomalies is likely more complex due to two-way interactions between tropical heating and the extratropics.

**2. Clarification and Explanation in Figure 1**

    a.  A significant amount of information is provided in the caption of Figure 1. Key details, such as the definition of the moist margin and the association of wet perturbations with weather objects, should be incorporated into the main text.

    b.  The section could benefit from a clearer explanation of what is depicted in Figure 1, such as explicitly stating how the MJO active object overlaps with the wet perturbation object and similar interactions for other objects.

    c.  Figure 1: The caption states, "The background moist margin, defined as the 90day running mean of 45 kg m$^{-2}$, is in black," whereas the main text mentions, "Wet perturbations are areas where the daily TCWV is above and where the background TCWV is below 45 kg m$^{-2}$." This appears contradictory. Clarify whether there is a distinction between the background moist margin and the background state, and define these terms in the text.

    a.  Although Figure 1 is introduced in the methods section (where objects are first introduced), we do not describe interactions between wet perturbations and weather objects until section 3.2. To clarify this, we refer to Section 3.2 for the part of the figure caption that refers to associations between objects. The definition of the background state moist margin is now made clearer in the caption and body text (see also response to comment 2c).

    b.  An example of MJO objects interacting with wet perturbations has been added to Section 3.2 (lines 151-152). Examples for LPS and PV anomalies are already provided in this section.

    c.  "Background moist margin" and "background state" refer to the same thing (the background moist margin is where the background state TCWV is 45 kg m$^{-2}$). We have clarified the meaning and used the term "background state" or "background state moist margin" consistently in the text and Figure 1 caption to clarify its meaning.

**3. Role of CCTWs in Wet Perturbations**

The discussion does not adequately explain why CCTWs only slightly modulate wet perturbations despite their significance in tropical rainfall variability. One possible explanation is that CCTWs have their highest amplitudes within the moist margin (e.g., see Figure 5 of Kiladis et al., 2009). Kelvin waves, in particular, have their greatest amplitude confined close to the equator, well within the moist margin. Thus, it is plausible that CCTWs primarily contribute to internal variability within the moist margin.

We agree with this comment and have added some discussion in lines 415-419 to reflect this.

It is worth noting that even though the modulation of the moist margin by CCTWs is small, is still statistically significant as shown in the new Section 3.3.

**4. Interactions of Multiple Weather Objects**

The manuscript does not address cases where more than one weather object contributes to a wet perturbation. For example, Kelvin waves and ER waves are often embedded in active MJO events, forming part of a larger weather event. Are such events double-counted in Figure 6 (panels c and d)? A figure similar to Figure 4 could be included to show events where MJO + Kelvin waves, MJO + ER waves, ER + Kelvin waves, and MJO + ER + Kelvin waves are associated with a wet perturbation. It could be valuable to test the hypothesis that simultaneous modes result in a greater effect on wet perturbations.

Wet perturbations may be associated with multiple weather objects, and as such are "double-counted" in Figure 6 along with other figures. For example, a wet perturbation associated with both MJO active and monsoon LPS weather systems will be counted in both categories.

We agree with the reviewer's comment suggesting that combined weather systems may result in a greater impact on wet perturbations. However, with nine types of weather objects analysed in the study, reproducing the maps in Figure 4 for all combinations of weather systems will result in far too much information to be easily digested.

As an alternative, we test the hypothesis that weather systems may combine for a greater effect by calculating some basic statistics of wet perturbations associated with multiple weather systems. Figure A3 displays the number, average object area, TCWV anomaly and daily precipitation of wet perturbations associated with two weather objects. Note that the latter three quantities correspond to the mean of the distributions shown in Figure 5 of the manuscript. The special case of MJO, ER and Kelvin all active is also shown in recognition that the MJO often contains ER and Kelvin wave structures.

[Figure]

Figure A3: Count (a), average object area (b), average TCWV anomaly (c) and average daily precipitation (d) of wet perturbations associated with two weather objects. The elements on the diagonal display values for objects associated with the single weather system. Here, the plus symbol refers to active components of CCTW objects and cyclonic PV anomalies, and the minus symbol refers to suppressed CCTWs and anticyclonic PV anomalies. The values for the special case of MJO, ER and Kelvin all active are shown in the box in the bottom left.

The results show that a substantial number of wet perturbations interact with at least two weather systems, as shown in Figure A1a. These objects tend to be larger, especially for those which are associated with both PV anomalies and CCTW objects (panel b). This is expected given CCTW objects are confined to near the equator and PV objects occur further poleward, meaning wet perturbations must be large to interact with both. The average TCWV anomaly tends to be larger for wet perturbations associated with multiple weather objects, although the differences are small (panel c). The main difference is evident in panel (d), which shows substantial modulation of rainfall when multiple weather objects are associated. Wet perturbations that are associated with active (suppressed) CCTW objects produce increased (decreased) rainfall regardless of other objects that are associated. For example, wet perturbations associated with all three active CCTWs produce more rainfall (7.22 mm/day) than those associated with MJO and ER active (7.00 mm/day), which is more than MJO active alone (6.18 mm/day). The wet perturbations that produce the most rainfall are those associated with both monsoon LPS and PV anomalies (10.38 mm/day).

It is possible that there is a dependence on object area underlying all of this. That is, are wet perturbations associated with multiple weather objects "stronger" simply because they are larger, and large objects are more likely to interact with multiple weather systems? We evaluate this by calculating the Pearson correlation coefficient between the quantities in Figure A3:

- Correlation between area and precipitation is 0.18
- Correlation between area and TCWV anomaly is 0.39
- Correlation between TCWV anomaly and precipitation is 0.30

This suggests that wet perturbations associated with multiple weather objects tend to have stronger TCWV anomalies due to their larger size. However, precipitation is more independent of size, and weather objects appear to play a clearer role.

Overall, these results suggest that multiple weather systems can indeed combine to result in a greater effect on the moist margin (mostly the rainfall). However, this effect is relatively small and seems to be somewhat linear with each weather system contributing independently.

We have added a brief paragraph in Section 3.5 (lines 330-334) to discuss the role of multiple weather systems on the moist margin.

**5. Indirect Influence of CCTWs on the Moist Margin**

As noted in Comment 3, CCTWs are potentially more important for the internal variability of TCWV within the moist margin. However, since they are responsible for high-amplitude rainfall in the tropics, they may release significant latent heat that causes upper-level divergence. This divergence could interact with the subtropical jet, producing PV anomalies. Hence the influence of CCTWs could be indirect. Investigating this could be a point of future work.

We agree with this comment and have added a sentence in the PV anomalies discussion (lines 435-436) as well as a sentence in the CCTW discussion paragraph (lines 417-419) to reflect this.

**Minor Comments**

1. **Introduction**
   a. Line 15: Consider rephrasing to "Observations reveal that the tropical atmosphere is largely characterised by..." Updated
2. **Section 2.2.2: Convectively Coupled Tropical Waves**
   a. Line 67: Clarify what is meant by the "20-degree latitude band." Does this refer to all latitudes within 20°S-20°N? If so, why does the Kelvin wave contour extend beyond 20°N? 20°S-20°N – this mistake has been fixed in the text. In both Figure 1 and 2, Kelvin wave objects (and more generally, CCTW objects) do not occur poleward of 20°S/N.
3. **Section 2.3**
   a. Line 107: The URL is not enclosed in brackets. Updated
4. **Section 3.1: Results**
   a. Clarify whether the objects in Figure 2 are only those overlapping with wet perturbations or all identified objects. If the latter, how does this change when considering only objects overlapping with wet perturbations? Figure 2 shows all identified objects, which has been clarified in the text (lines 130-131). The number of wet perturbations that interact with weather objects is shown below in Figure A4. Note that all lower panels largely resemble the total frequency of wet perturbations at the top. Therefore, we consider the ratio of interacting wet perturbations (lower panels of Fig A3) with the total number of wet perturbations (top panel of Fig A3) to be a more useful indication of the role of weather systems on the moist margin. These "proportions" (also see response to minor comment 5a) are presented in Figure 4 of the manuscript, and for brevity and clarity we choose to exclude the results in Figure A1 below.

[Figure]

Figure A4: total count of wet perturbations (top) and wet perturbations associated with each weather object. Note the different colour scale in the top panel.

**5. Section 3.2**

    a. Figure 3: Define "proportion." Is it the ratio of objects during wet perturbations? Note in the caption that proportions add to a value greater than one, possibly due to multiple objects contributing to a wet perturbation (related to Major Comment 4).

"Proportion" here means the number of wet perturbations associated with a weather object divided by the total number of wet perturbations. This has been added in the text (lines 164-165). We have also added a comment in the caption explaining that proportions sum to greater than one as suggested.

    b.  Figure 4: The caption mentions numbers in brackets, but these are absent from the panels.

This part of the caption has been removed.

**6. Section 3.3**

    a.  Figure 6: The caption needs clarity. Could the authors explain how the composites are "relative" to the centre of mass and what "by their associated weather objects" means? Also, clarify "pseudo-latitude" and its necessity. Perhaps replace it with "poleward" for better comprehension. Additionally, explain why negative pseudo-latitude is used and what "pseudo-longitude" means in the caption.

We have reworded some of the text explaining Figure 6 for greater clarity.

- "Relative" has been changed to in the region surrounding"
- "...by their associated weather objects" has been changed to "...that are associated with different weather objects" (lines 246-247), and similarly in the caption of Figure 5
- pseudo-latitude and pseudo-longitude coordinates have been explained in the text as relative to the centre of mass of the wet perturbation object (lines 248-249)
- We decide to keep "pseudo latitude" since "poleward latitude" is not consistent with "pseudo longitude" and "pseudo poleward latitude" seems overly verbose.
- Negative pseudo latitude simply refers to the equatorward direction relative to objects – this should be clear from lines 253-254.

    b.  Line 216: The text states, "There exists a wave-like pattern that propagates equatorward towards the east." As this is a spatial composite, propagation cannot be inferred. Perhaps rephrase to indicate the pattern slants eastward.

The word "propagates" has been changed to "slants" to avoid confusion.

**7. Section 4.1**

    a. Line 290: "A wet perturbation event is said to be associated with a weather object if the conditions in the previous section are satisfied at any point..." Restate these conditions for clarity.

This has been added into the text in brackets (lines 349-350)